# Differential accumulation of storage bodies with aging defines discrete subsets of microglia in the healthy brain

Jeremy Carlos Burns[1,2], Bunny Cotleur[3], Dirk M Walther[4], Bekim Bajrami[4], Stephen J Rubino[1], Ru Wei[4], Nathalie Franchimont[5], Susan L Cotman[6], Richard M Ransohoff[7], Michael Mingueneau[1]*

[1]Multiple Sclerosis & Neurorepair Research Unit, Biogen, Cambridge, United States; [2]Department of Pharmacology & Experimental Therapeutics, Boston University School of Medicine, Boston, United States; [3]Emerging Neurosciences Research Unit, Biogen, Cambridge, United States; [4]Chemical Biology and Proteomics, Cambridge, United States; [5]Multiple Sclerosis Development Unit, Biogen, Cambridge, United States; [6]Center for Genomic Medicine, Department of Neurology, Massachusetts General Hospital, Harvard Medical School, Boston, United States; [7]Third Rock Ventures, Boston, United States

**Abstract** To date, microglia subsets in the healthy CNS have not been identified. Utilizing autofluorescence (AF) as a discriminating parameter, we identified two novel microglia subsets in both mice and non-human primates, termed autofluorescence-positive ($AF^+$) and negative ($AF^-$). While their proportion remained constant throughout most adult life, the AF signal linearly and specifically increased in $AF^+$ microglia with age and correlated with a commensurate increase in size and complexity of lysosomal storage bodies, as detected by transmission electron microscopy and LAMP1 levels. Post-depletion repopulation kinetics revealed $AF^-$ cells as likely precursors of $AF^+$ microglia. At the molecular level, the proteome of $AF^+$ microglia showed overrepresentation of endolysosomal, autophagic, catabolic, and mTOR-related proteins. Mimicking the effect of advanced aging, genetic disruption of lysosomal function accelerated the accumulation of storage bodies in $AF^+$ cells and led to impaired microglia physiology and cell death, suggestive of a mechanistic convergence between aging and lysosomal storage disorders.

*For correspondence:
michael.mingueneau@biogen.com

## Introduction

Microglia are a unique population of tissue resident macrophages residing in the central nervous system (CNS) accounting for 10% to 15% of all cells within the CNS. While displaying some canonical macrophage activities such as the phagocytosis of debris and apoptotic bodies (*Chan et al., 2001*; *Janda et al., 2018*), microglia are also endowed with functions specific to the CNS microenvironment (*Clayton et al., 2017*; *Li and Barres, 2018*; *Ransohoff, 2016*; *Ransohoff and Khoury, 2016*), such as synaptic remodeling (*Paolicelli et al., 2011*; *Stephan et al., 2012*; *Stevens et al., 2007*; *Weinhard et al., 2018*), neuronal support (*Parkhurst et al., 2013*; *Ueno et al., 2013*), and oligodendrogenesis (*Hagemeyer et al., 2017*; *Wlodarczyk et al., 2017*).

However, despite this diversity of functions, no durable subsets have been identified in the healthy adult brain at steady-state. The disease-associated microglia (DAM) and the closely-related microglia subset expressing a neurodegenerative phenotype (MGnD) were reported to arise in response to the accumulation of β-amyloid plaques in Alzheimer's Disease (AD) transgenic mouse models (*Kamphuis et al., 2016*; *Keren-Shaul et al., 2017*; *Krasemann et al., 2017*; *Mrdjen et al., 2018*), as well as in other models of neurodegeneration and aging (*Chiu et al., 2013*;

**eLife digest** Microglia are a unique type of immune cell found in the brain and spinal cord. Their job is to support neurons, defend against invading microbes, clear debris and remove dying neurons by engulfing them. Despite these diverse roles, scientists have long believed that there is only a single type of microglial cell, which adapts to perform whatever task is required. But more recent evidence suggests that this is not the whole story.

Burns et al. now show that we can distinguish two subtypes of microglia based on a property called autofluorescence. This is the tendency of cells and tissues to emit light of one color after they have absorbed light of another. Burns et al. show that about 70% of microglia in healthy mouse and monkey brains display autofluorescence. However, about 30% of microglia show no autofluorescence at all. This suggests that there are two subtypes of microglia: autofluorescence-positive and autofluorescence-negative.

But does this difference have any implications for how the microglia behave? Autofluorescence occurs because specific substances inside the cells absorb light. In the case of microglia, electron microscopy revealed that autofluorescence was caused by structures within the cell called lysosomal storage bodies accumulating certain materials. The stored material included fat molecules, cholesterol crystals and other substances that are typical of disorders that affect these compartments. Burns et al. show that autofluorescent microglia contain larger amounts of proteins involved in storing and digesting waste materials than their non-autofluorescent counterparts. Moreover, as the brain ages, lysosomal storage material builds up inside autofluorescent microglia, which increase their autofluorescence as a result. Unfortunately, this accumulation of cellular debris also makes it harder for the microglia to perform their tasks.

Increasing evidence suggests that the accumulation of waste materials inside the brain contributes to diseases of aging. Future work should examine how autofluorescent microglia behave in animal models of neurodegenerative diseases. If these cells do help protect the brain from the effects of aging, targeting them could be a new strategy for treating aging-related diseases.

---

*Holtman et al., 2015*; *Spiller et al., 2018*; *Wlodarczyk et al., 2014*). Mirroring the gene signature and function of DAM and MGnD subsets in disease conditions, the proliferative-region-associated microglia (PAM) subset was associated with the phagocytosis of newly formed and dying oligodendrocytes during normal post-natal development (*Felsky et al., 2019*; *Hagemeyer et al., 2017*; *Li et al., 2019*). Dark microglia, a subset identified by their condensed, electron-dense cytoplasm visible by transmission electron microscopy, are frequently observed during pathologic states and believed to actively associate with neuronal synapses (*Bisht et al., 2016*). Beyond conditions of neuronal or oligodendrocyte cell death, microglia heterogeneity was also observed within the adult steady-state CNS. In particular, regional variations in cellular density (*Lawson et al., 1990*), but also in microglia gene expression profiles were reported, pointing to region-specific microglia functions and possibly subsets in healthy brain (*Ayata et al., 2018*; *Grabert et al., 2016*).

While these emerging transcriptomic findings suggest that steady-state microglia subsets are likely present, they remain to be identified and characterized. By using cellular autofluorescence (AF) as a novel photophysical parameter to explore microglia heterogeneity in unperturbed conditions, we report that steady-state microglia exist in two discrete states at a regulated ratio throughout the entire lifespan of rodent and non-human primate species. Leveraging this physical property, we devised a novel probe-free method to isolate and characterize these subsets and established that $AF^+$ and $AF^-$ microglia differed in their ultrastructural features, homeostatic dynamics, proteomic content and physiological properties.

## Results

### Cellular autofluorescence identifies two discrete microglia subsets

Based on the observation that certain peripheral myeloid populations are highly autofluorescent, we initially included an empty fluorescence channel in our microglia flow cytometry analyses. Doing so revealed an unexpected bimodal distribution of autofluorescence (AF) intensity in $CD45^{dim}CD11B^+$

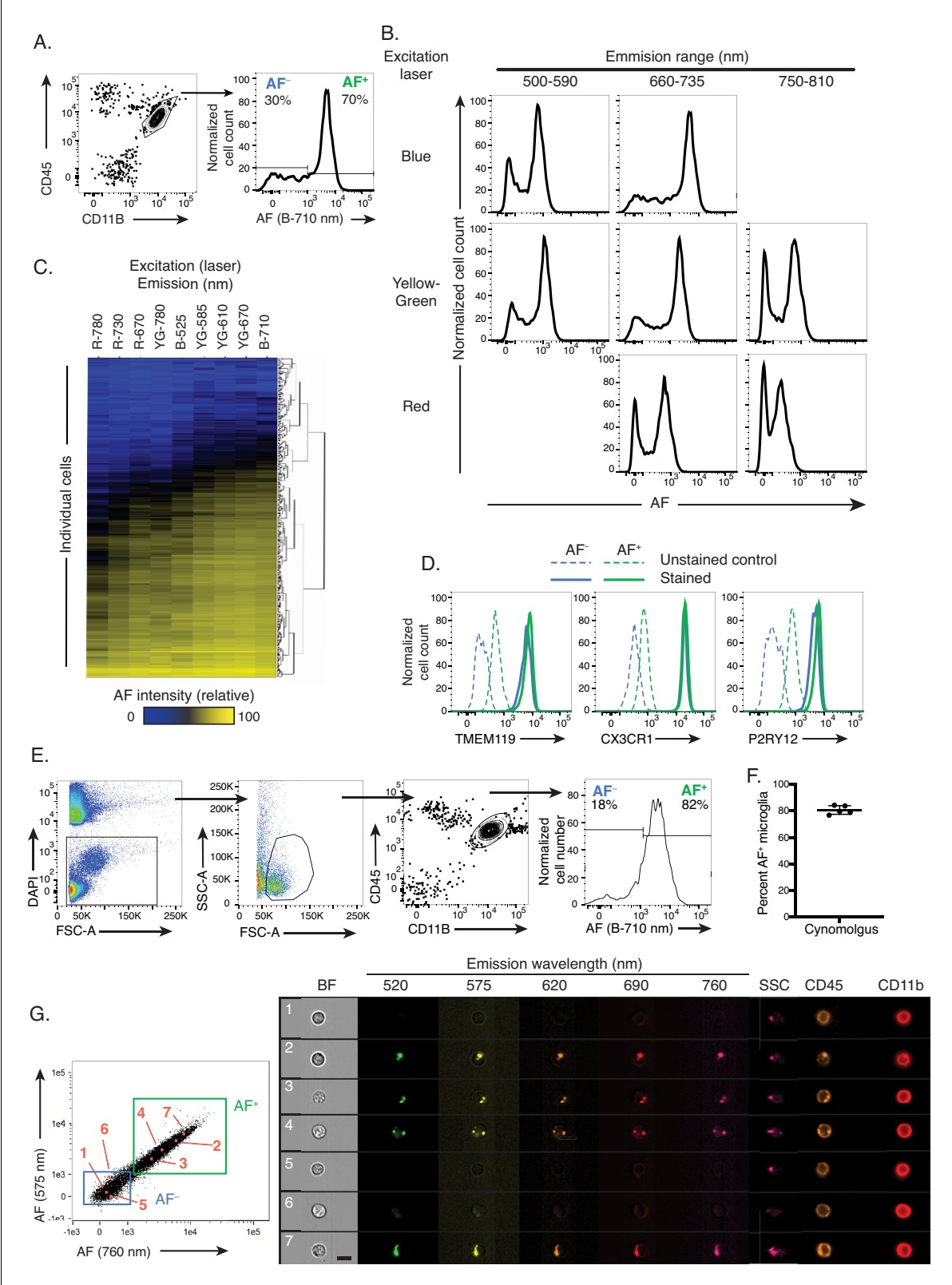

**Figure 1.** Cellular autofluorescence identifies two discrete and novel microglia subsets. (**A**) Representative flow cytometry scatterplot showing CD45 and CD11B surface levels in a brain single cell suspension and histogram of autofluorescence (AF) signal in DAPI⁻ CD45^dim CD11B⁺ microglia detected with a blue laser and 710 nm detector. (**B**) Representative flow cytometry histograms of AF intensity in the entire microglia population for multiple excitation lasers and emission filters. (**C**) Heatmap of AF signal detected across multiple cytometer channels (columns) in 500 single microglial cells

*Figure 1 continued on next page*

*Figure 1 continued*

(rows) and hierarchically clustered by Manhattan distance. (D) Representative flow cytometry histograms of TMEM119, CX3CR1 and P2RY12 surface levels in AF$^+$ and AF$^-$ microglia subsets. (E) Representative flow cytometry scatterplots used to identify microglia in brain single cell suspensions from Cynomolgus macaques and representative histogram of AF levels detected in microglia. (F) Quantitation of results presented in (E). (G) Representative scatterplots of AF levels detected in CD45$^{dim}$CD11B$^+$ microglia and corresponding microscopy images of selected AF$^+$ and AF$^-$ cells (highlighted in orange and numbered in the scatterplots and images) analyzed using imaging flow cytometry. Scale bar = 7 μm. (A, B) Representative of n = 12 animals from at least four independent experiments. (D) n = 6 animals from at least two independent experiments. (E, F) n = 5 animals from two independent experiments. B = blue laser, R = red laser, YG = yellow green; AF = autofluorescence; BF = bright field; SSC = side scatter. See also *Figure 1—figure supplement 1*.

The online version of this article includes the following figure supplement(s) for figure 1:

**Figure supplement 1.** Autofluorescence identifies two discrete microglia subsets across brain regions and in both male and female mice.

microglia isolated from 6-month-old naïve C57BL/6 mice (*Figure 1A* and *Figure 1—figure supplement 1A*). This bimodal distribution identified two subsets of microglia: an AF-positive (AF$^+$) subset showing a strong AF signal, and an AF-negative (AF$^-$) subset which displayed no or minimal levels of AF. The two subsets appeared at a highly reproducible ratio of 1:2.5 (AF$^-$:AF$^+$), with an average frequency of AF$^+$ microglia of 71 ± 1.2% (n = 13; CV = 1.6%) (*Figure 1—figure supplement 1B*). To further characterize the spectral properties of the AF signal, we included additional empty fluorescence channels. The AF signal showed maximal intensity in the 660–735 nm emission range upon excitation with a 488 nm (Blue) laser but was also detected across multiple combinations of laser excitation wavelengths and emission filter ranges (*Figure 1B*). Extraction of single cell-level data from flow cytometry analyses revealed that the majority of microglia that were either positive or negative for AF in the Blue-710 nm AF channel were also positive or negative across most other AF channels tested, respectively (*Figure 1C*). In a few channels such as Red-780 nm for instance, 10% to 20% of the cells that were positive in Blue-710 nm appeared low or negative, which likely resulted from a differential sensitivity of the channels used to detect AF as shown in *Figure 1B* by the differential brightness of the AF$^+$ population in those two channels. Both AF microglia subsets were detected across different regions of the brain and while there were minor differences in the frequency of AF$^+$ microglia between regions, AF$^+$ cells displayed similar levels of AF signal in cerebellum, cortex and hippocampus (*Figure 1—figure supplement 1C,D*). Both AF$^+$ and AF$^-$ subsets of microglia were positive for microglia homeostatic markers, including CX3CR1, P2RY12 and TMEM119 (*Figure 1D* and *Figure 1—figure supplement 1E*). Finally, there were no gender differences observed in the frequency of AF$^+$ microglia nor the intensity of AF (*Figure 1—figure supplement 1F,G*). This observation was conserved across species as microglia isolated from the brains of 3- to 4-year-old Cynomolgus monkeys showed a very consistent bimodal pattern of AF in the Blue-710 nm channel with 80 ± 3.6% of microglia that were AF$^+$ on average (*Figure 1E,F*). Altogether, these analyses identified cellular AF as a novel photophysical property for discriminating two discrete subsets of microglia present at steady-state and conserved between rodents and non-human primates.

## The autofluorescence signal in AF$^+$ microglia originates from intracellular organelles

To define the subcellular origin of the AF signal in microglia, we utilized imaging cytometry. Microscopy images of AF$^+$ cells identified by the fluorescence intensity detected in two empty channels (*Figure 1G*) revealed that the AF signal was not diffusely distributed throughout the cellular volume but was punctate and localized within intracellular organelles. The subcellular AF compartments were observed in all tested AF channels and systematically colocalized (*Figure 1—figure supplement 1H and I*). AF$^-$ cells did not display detectable AF subcellular compartments in any of the tested channels, altogether establishing that these two subsets of microglia, identified solely by their AF profiles, differed by the presence of highly autofluorescent intracellular organelles that were restricted to the AF$^+$ microglia subset.

## AF signal intensity increases linearly with aging, but solely within AF$^+$ microglia

In contrast to the punctate AF signal detected in 3-month-old animals, imaging flow cytometry applied to naïve mice from a range of ages revealed that AF subcellular structures became largely

confluent in 10-month-old animals and occupied a larger fraction of the cytosol (from 11 µm$^2$ to 17 µm$^2$ on average in 3- and 10- month-old animals, respectively) (*Figure 2A,B*). Consistent with these results, flow cytometry analyses established a nearly linear increase of AF signal intensity in AF$^+$ microglia with aging (*Figure 2C,D*), resulting in a cumulative 3-fold increase of AF signal between 3- and 12-month-old mice across multiple fluorescence channels (*Figure 2D* and *Figure 2—figure supplement 1A*). While a clear AF$^+$ population was not detectable in mice aged 15 days post-natal or younger, a bimodal distribution appeared as early as 30 days post-natal and AF increased linearly with age from that point on (*Figure 2—figure supplement 1B,C*). In contrast, age-dependent

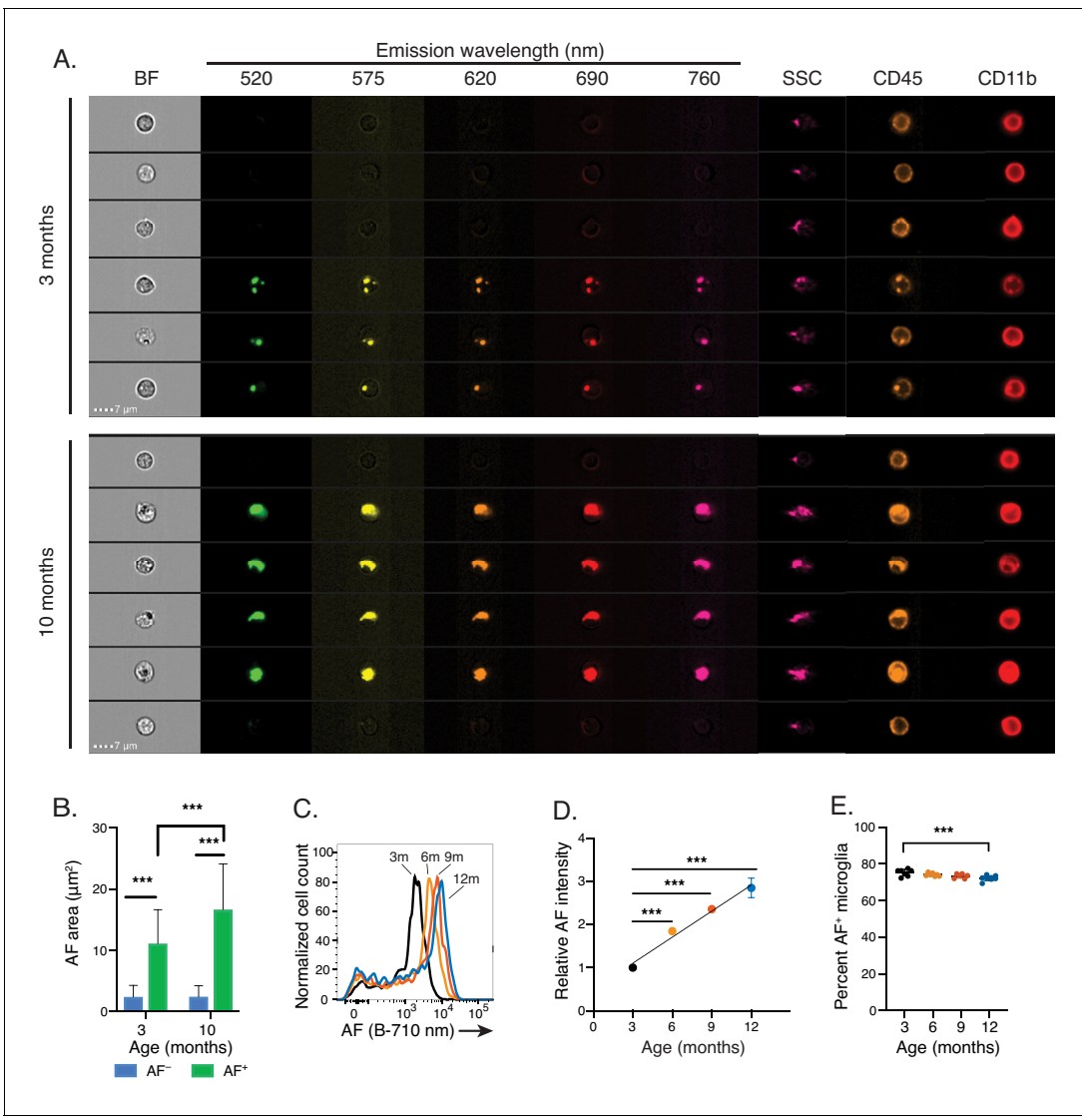

**Figure 2.** Natural aging increases microglia AF signal intensity solely within AF$^+$ cells. (**A**) Representative imaging flow cytometry images and (**B**) corresponding quantitation of average AF area per cell in AF$^+$ and AF$^-$ CD45$^{dim}$CD11B$^+$ microglia isolated from 3- and 10-month old mice. Significance established with 2-way ANOVA followed by Sidak's post-hoc correction, n = 4000–8000 cells. (**C**) Representative flow cytometry histogram and (**D**) corresponding quantitation of AF intensity in AF$^+$ CD45$^{dim}$ CD11B$^+$ microglia isolated from mice at the indicated ages and presented as normalized values to AF levels observed in 3 month old animals. Significance established with 1-way ANOVA followed by Dunnett's post-hoc. The line depicts the linear regression (R$^2$ = 0.95). (**E**) Percent AF$^+$ microglia at indicated ages. Significance established with 1-way ANOVA followed by Dunnett's post-hoc. All data are presented as mean ± SD. For panels C-E, n ≥ 6 animals per age from at least two independent experiments. BF = bright field, AF = autofluorescence, SSC = side scatter, ***p<0.001. See also *Figure 2—figure supplement 1*.

The online version of this article includes the following figure supplement(s) for figure 2:

**Figure supplement 1.** Microglial autofluorescence increases with age in multiple flow cytometry channels and is unique to microglia.

increases in cellular AF were not detected in peripheral CD11B$^+$ cells residing in the spleen (*Figure 2—figure supplement 1D*). Despite the large increase in the AF signal observed in AF$^+$ cells, their frequency remained largely unchanged during aging, decreasing only slightly from 75% to 72% from 3 to 12 months of age (*Figure 2E*). Finally, neither the proportion of AF$^-$ cells (*Figure 2E*) nor the area of intracellular AF signal detected in AF$^-$ microglia (*Figure 2A,B*) were altered by aging, revealing a selective impact of aging on the AF$^+$ microglia subset.

## AF$^+$ microglia selectively accumulate intracellular storage bodies with age

We next isolated AF$^+$ and AF$^-$ microglia from 3- and 18-month-old mice using fluorescence-activated cell sorting (FACS) and performed transmission electron microscopy (TEM) analyses. At 3 months of age, the intracellular organization of AF microglia subsets differed in that AF$^+$ cells almost systematically contained large storage bodies filled with osmophilic electron-dense deposits (*Figure 3A*). These electron-dense storage bodies were observed in 80% of AF$^+$ microglia whereas only 46% of AF$^-$ microglia contained electron-dense organelles (*Figure 3A,C*). When observed in AF$^-$ cells, storage bodies were devoid of complex storage material and were more regularly shaped than those observed in AF$^+$ cells (*Figure 3A*). In aged mice, the storage bodies within AF$^+$ microglia changed dramatically in both size and complexity (*Figure 3A*). Between 3 and 18 months of age, the percentage of visible cytoplasm occupied by storage bodies increased from 9% to 23% on average and approximately 30% of AF$^+$ cells from aged mice contained AF material that occupied more than a third of the cytoplasm (*Figure 3D*). The ultrastructural composition of the storage bodies in AF$^+$ cells varied with frequent curvilinear (black arrowheads) and fingerprint-like profiles (red arrowheads) (*Figure 3B*). White, linear, rod-like material with fine-tipped ends (white arrowheads) were observed within membrane-bound lipoid bodies and closely resembled cholesterol crystal deposits (*Figure 3B*). Most prominent, however, was the proportion and volume of lipid droplets (asterisks) (*Figure 3B*). In contrast, both the frequency of cells containing storage bodies and the proportion of the cytosol occupied by storage bodies remained unchanged in the AF$^-$ microglia subset (*Figure 3A,C,D*).

In addition to these ultrastructural differences, AF$^+$ cells expressed higher levels of LAMP1 and CD68 compared to AF$^-$ cells (*Figure 3E,F*), indicating an enlargement of endolysosomal storage compartments in AF$^+$ cells. Furthermore, a gradual age-dependent increase in LAMP1 and CD68 protein levels was observed in the AF$^+$ subset (*Figure 3E,F*) whereas AF$^-$ microglia did not show changes. Altogether these observations indicated that AF$^+$ microglia differed from AF$^-$ cells by their unique accumulation of endolysosomal storage compartments with aging.

## Microglia AF subsets exhibit differential population dynamics upon depletion and replenishment of the microglia niche

To explore the cellular dynamics and ontogenic relationships between these two novel subsets of microglia, we depleted microglia in 14-month-old mice with the CSF1R-small molecule antagonist BLZ945 (*Krauser et al., 2015*; *Pyonteck et al., 2013*). Twenty-four hours following the treatment period, depletion was nearly complete, as assessed by the remaining frequency and absolute numbers of microglia (*Figure 4A,B* and data not shown). Consistent with previous reports (*Huang et al., 2018*; *O'Neil et al., 2018*), microglia rapidly repopulated the CNS at 7 and 14 days post-treatment, recovering on average to 25% (at 7 days) and 87% (at 14 days) of steady-state microglia numbers (*Figure 4A* and data not shown). However, repopulation by AF$^+$ and AF$^-$ subsets showed distinct kinetics. While AF$^-$ microglia cell numbers reached 67% of steady-state levels by day 7, AF$^+$ cells only repopulated to 4% of steady-state levels by that time (*Figure 4B*). At 14 days repopulating AF$^+$ cell numbers only reached 14% of steady-state AF$^+$ values in vehicle-treated mice while the repopulating AF$^-$ subset surpassed steady-state levels by approximately 2-fold (*Figure 4B*) before normalizing to steady-state levels at day 70. Altogether these results established that the AF$^-$ subset was the first subset to repopulate the depleted brain while the repopulation by AF$^+$ cells was delayed, suggesting a possible conversion from AF$^-$ to AF$^+$ state during repopulation.

Supporting this hypothesis, the AF intensity of the repopulated microglia slowly increased from barely detectable levels at day seven to slightly increased levels at day 14, ultimately returning to a bimodal distribution by day 70 (*Figure 4C,D*). Even at the latter timepoint, only a small fraction of

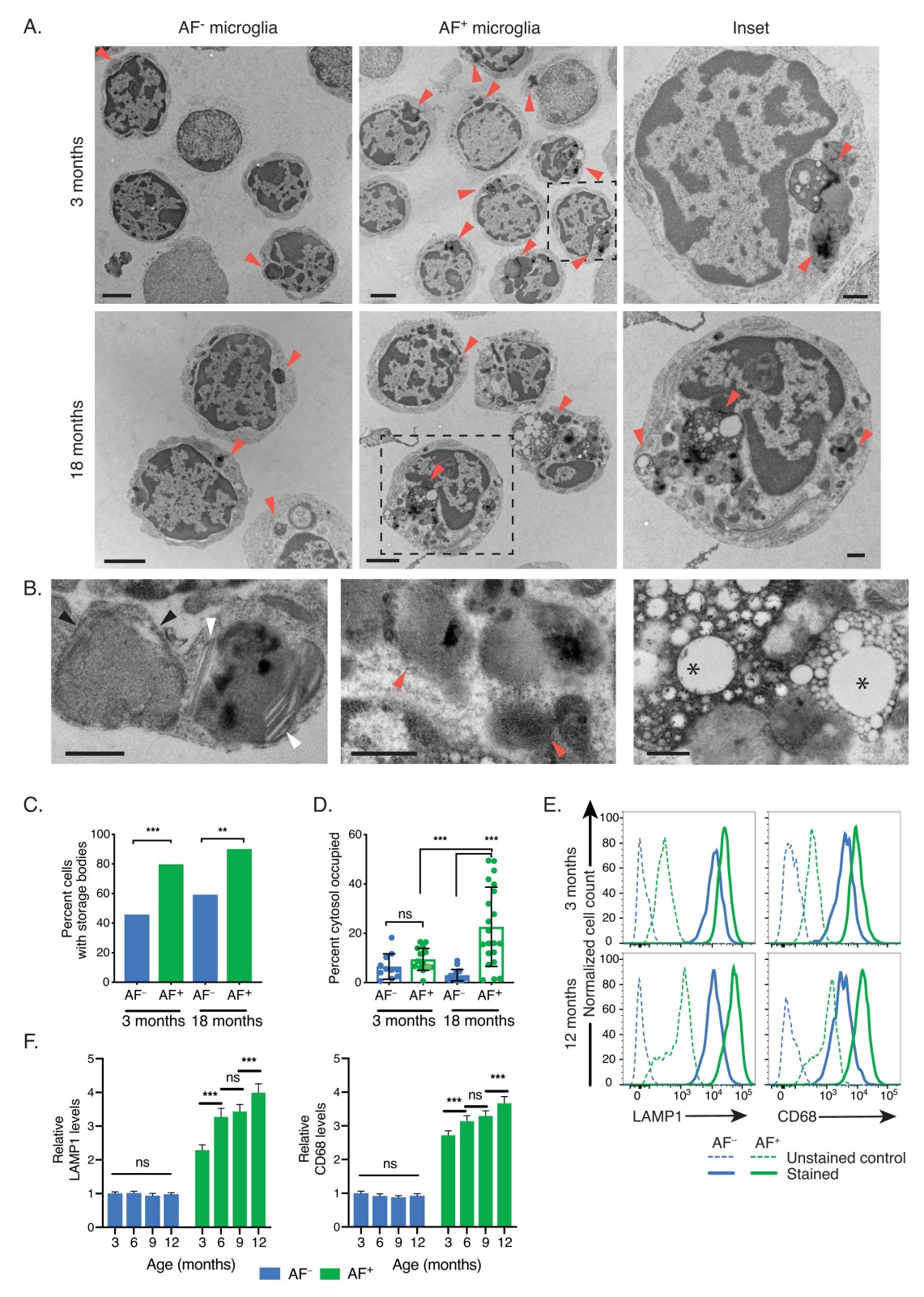

**Figure 3.** AF+ microglia selectively accumulate intracellular storage bodies with age. (**A**) Representative TEM images of sorted microglia subsets from 3- to 18-month-old mice. Red arrowheads point to storage bodies. Scale bars = 2 μm. Insets depict higher magnification images of selected AF+ microglia (dashed square lines). Inset scale bars = 500 nm. (**B**) Representative TEM images of storage bodies observed within aged AF+ microglia. Black arrowheads, curvilinear storage material; white arrowheads, crystal bodies; red arrows, fingerprint-like storage material; asterisks, lipid storage content.
*Figure 3 continued on next page*

Figure 3 continued

Scale bars = 500 nm. (C) Frequency of cells containing storage bodies in microglia AF subsets sorted and pooled from n = 10 mice at the indicated ages. Significance established with Fischer's exact test. n = 48–59 cells/group. (D) Average percent of cytosolic area occupied by storage material in cells containing at least one storage body. Significance established with 2-way ANOVA followed by Tukey's post-hoc. n = 11–21 cells/group. (E) Representative flow cytometry histograms and (F) corresponding quantitation of LAMP1 and CD68 staining in AF$^-$ and AF$^+$ subsets at indicated ages. Relative levels of LAMP1 and CD68 calculated as net geometric mean fluorescence intensity after subtraction of background AF signal and normalization to AF intensity detected in the AF$^-$ subset at 3 months of age. Significance established with 2-way repeated-measures ANOVA followed by Tukey's post-hoc. n ≥ 6 animals per age from at least two independent experiments. All data presented as mean ± SD. ns = not significant, **p<0.01 ***p<0.001.

this newly-formed AF$^+$ subset (19% on average) displayed AF intensity levels comparable to those seen in vehicle-treated mice (*Figure 4C,D* and *Figure 4—figure supplement 1A,B*) while most repopulating AF$^+$ cells displayed 35% weaker AF signal intensity on average. This gradual accumulation of AF material over time in repopulating microglia supported the possibility that AF$^+$ microglia were derived from the conversion of AF$^-$ cells during replenishment. Further validating this conclusion, 56% of microglia displayed positivity for the proliferation-associated marker KI-67 at day seven before returning to steady-state values by day 14 (*Figure 4E,F*). Because AF$^+$ microglia were virtually absent during this early repopulation phase (*Figure 4B*), these proliferation kinetics implied that the AF$^-$ subset of microglia was the predominant subset responsible for the repopulation of the microglia compartment following depletion and that AF$^+$ microglia were derived from the conversion over time of a subset of AF$^-$ microglia.

## Proteomic analysis of isolated AF$^+$ and AF$^-$ microglia subsets reveals molecular differences in endolysosomal, autophagic and metabolic pathways

To determine whether microglia AF subsets were distinct at the molecular level, FACS-isolated AF$^+$ and AF$^-$ microglia were analyzed by nano liquid chromatography mass spectrometry (nLC-MS). A total of 4231 proteins were detected by label-free LC-MS/MS and after filtering for proteins quantified in at least 50% of samples in either subset, 3492 remained for analysis. 351 proteins showed significant differences in abundance between AF$^+$ and AF$^-$ microglia (Benjamini-Hochberg adj. p value < 0.01, fold-change > |1.3|), with 254 and 97 upregulated and downregulated differentially expressed proteins (DEPs), respectively (*Figure 5A* and *Figure 5—source data 1*). When ranked by significance, 32 of the top 50 DEPs upregulated in AF$^+$ microglia were associated with endolysosomal biology, among which was LAMP1, validating prior flow cytometry results (*Figure 3E,F*). Furthermore, a large number of lysosomal enzymes were upregulated in AF$^+$ microglia including cathepsins (CTSA, CTSB, CTSD, CTSF, CTSL, CTSZ) as well as several enzymes involved in lysosomal degradation such as amidases (ASAH1, GBA, NAAA), thioesterases (PPT1, PSAP, NAGA), proteases (TPP1, LGMN) and glycosyl hydrolases (HEXA, HEXB, GLB1). Many other DEPs involved in the biology, trafficking, or fusion of endosomes with lysosomes and phagosomes (ATP6V0D1, SCARB2, GRN, ARL8B, TOM1, STX7, TMEM55B) were also upregulated in AF$^+$ microglia. Underscoring the importance of these proteins towards maintaining proper CNS homeostasis, genetic perturbations in 15 of the top 50 DEPs were associated with severe CNS-related storage disorders, such as Neuronal Ceroid Lipofucinosis and Niemann-Pick. Lastly, DEPs that are typically associated with neurons (CBLN1, CBLN4, SNAP25), oligodendrocytes (MOBP, SYNE2) and astrocytes (GJA1) were detected at significantly higher levels in AF$^+$ microglia, suggesting functional differences in either the phagocytic capacity of this subset or its ability to fully degrade ingested material.

To systematically explore additional pathways distinguishing AF subsets at the molecular level, Gene Ontology (GO) term enrichment was performed with Panther (*Figure 5B* and *Figure 5—source data 2*). DEPs upregulated in AF$^+$ microglia were enriched in pathways related to intracellular vesicle-mediated transport, lysosomal organization, protein transport, lipid catabolic processes and regulation of TOR signaling (*Figure 5B,C*). DEPs downregulated in AF$^+$ microglia showed enrichment in RNA-related biological processes (transcription, RNA metabolism, splicing) and chromatin silencing. In agreement with GO term enrichment analysis, the top canonical pathways identified by Ingenuity Pathway Analysis (IPA) as enriched in AF$^+$ microglia included phagosome maturation, autophagy, numerous catabolic pathways (amino acid and glutaryl-coA catabolism, ketogenesis and

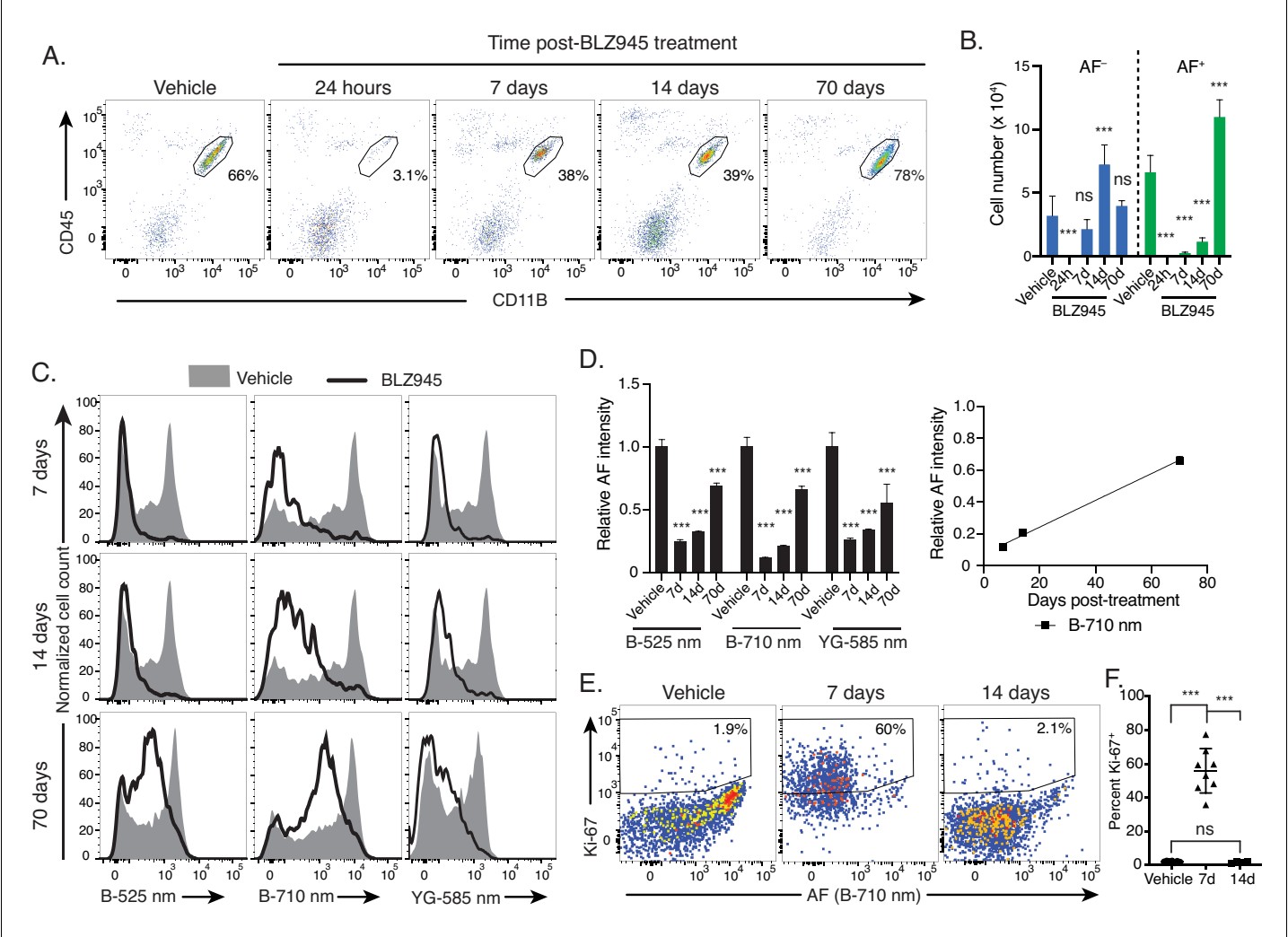

**Figure 4.** Microglia AF subsets exhibit differential population dynamics upon depletion and replenishment of the microglia niche. (A) Representative flow cytometry scatterplots showing CD45 and CD11B surface levels in brain single cell suspensions from mice at indicated timepoints post treatment with vehicle or BLZ945 and (B) corresponding quantitation of absolute cell numbers of AF$^+$ and AF$^-$ microglia post-treatment. Significance established with 1-way ANOVA followed by Dunnett's post-hoc test. (C) Representative flow cytometry histograms and (D) corresponding quantitation of AF intensity in the entire microglia population detected across multiple combinations of excitation lasers and emission filters and normalized to levels observed in vehicle-treated mice. In the scatter plot, the line represents the linear regression ($R^2 = 0.99$). Significance established with 1-way ANOVA followed by Dunnett's post-hoc test for each AF channel. (E) Representative flow cytometry scatterplots depicting KI-67 levels and AF intensity in microglia isolated at indicated time points post-treatment and (F) corresponding quantitation of the percent microglia positive for KI-67. Significance established with Welch's ANOVA followed by Dunnett's T3 post-hoc. $n \geq 8$ animals per genotype group from at least two independent experiments. All data represented as mean ± SD. d = day; h = hour; B = blue laser; YG = yellow green laser; AF = autofluorescence; ns = not significant, ***p<0.001. See also *Figure 4—figure supplement 1*.

The online version of this article includes the following figure supplement(s) for figure 4:

**Figure supplement 1.** Repopulating microglia exhibit lower levels of AF.

fatty acid β-oxidation), mitochondrial dysfunction, and pathways that were indicative of mTOR deregulation (mTOR, AMPK) (*Figure 5D* and *Figure 5—source data 3*). Finally, upstream regulators predicted to explain the distinct proteome displayed by AF$^+$ cells included transcription factors involved in the regulation of cell cycle, senescence and apoptosis (TP53, MYC, CDKN2A), ER stress and unfolded protein response (XBP1), autophagy and lysosomal biogenesis (TFEB) and inflammatory responses (NFKBIA, PPARA) (*Figure 5E* and *Figure 5—source data 3*). Altogether, these results established that AF$^+$ microglia expressed a unique proteome characterized by an increased representation of endolysosomal, autophagic, catabolic, and mTOR-related proteins.

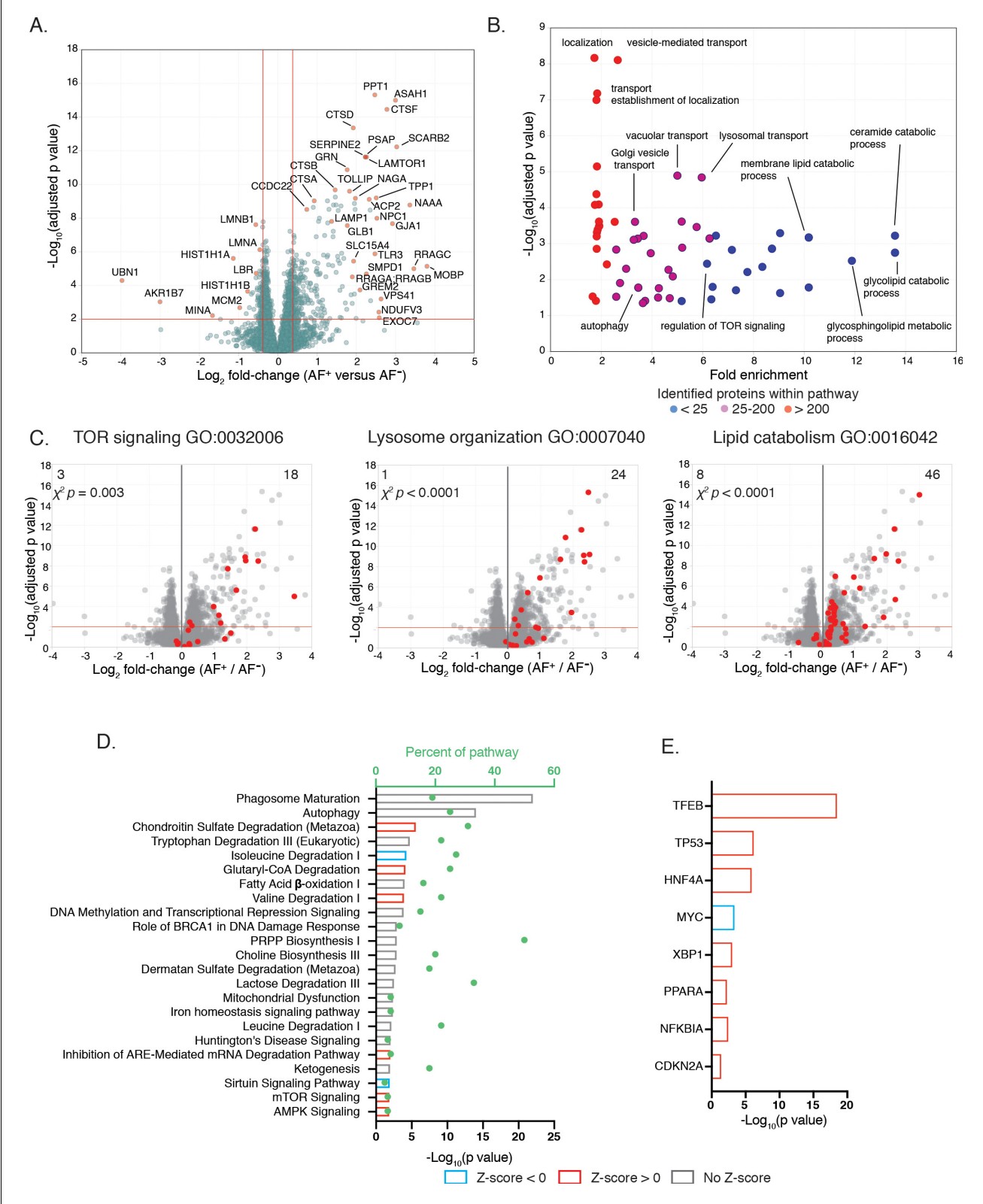

**Figure 5.** Proteomic analysis of isolated AF⁺ and AF⁻ microglia subsets reveals molecular differences in endolysosomal, autophagic and metabolic pathways. (**A**) Volcano plot comparing protein abundance in AF⁺ and AF⁻ microglia (x-axis: $Log_2$ abundance difference, y-axis: negative log Benjamini-Hochberg adjusted p value from two-tailed Student's t-test) with most-differentially expressed proteins annotated. Red lines: significance cutoffs (adjusted p value < 0.01, |Fold change| > 1.3). (**B**) Scatter plot displaying adjusted p value and GO term fold-enrichment results among differentially

*Figure 5 continued on next page*

*Figure 5 continued*
expressed proteins (DEPs) upregulated in AF$^+$, with selected pathways highlighted. Dot color indicates number of proteins within GO pathway detected in the dataset (see legend). (C) Volcano plots comparing protein levels in AF$^+$ and AF$^-$ microglia with indicated GO term signatures overlaid in red. p value, $\chi^2$ statistical test for bias in distribution of GO term pathway members. (D) Top canonical pathways and (E) upstream regulators identified by IPA as differentially regulated between AF subsets. (D) Bars and dots respectively indicate p values and the percent of pathway proteins detected. (D, E) Predicted activation or inhibition of pathway or transcriptional regulator is indicated by positive and negative Z-scores, respectively, see legend. Analysis based on 351 DEPs identified using adjusted p value < 0.01 and fold change > |1.3|. See also *Figure 5—source data 1*, *2* and *3*.
The online version of this article includes the following source data for figure 5:

**Source data 1.** AF$^+$ versus AF$^-$ subset associated proteins.
**Source data 2.** Panther overrepresentation test of AF$^+$ and AF$^-$ associated proteins.
**Source data 3.** Ingenuity pathway analysis of differentially regulated proteins in AF subsets.

## Myelin, Fc receptor-mediated and TREM2-mediated phagocytosis are not dominant mechanisms contributing to microglia AF accumulation

Given the abundance of intracellular storage bodies and the increased detection of proteins involved in endocytosis and phagosome maturation observed in AF$^+$ microglia, we investigated whether differential phagocytosis of myelin debris contributed to the subset-specific accumulation of AF storage material. Shiverer mice (*Mbp*$^{shi/shi}$) harbor a spontaneous autosomal-recessive mutation in myelin basic protein (MBP) which results in defective myelin compaction and physical instability of the myelin sheath (*Privat et al., 1979*; *Weil et al., 2016*). Despite this myelin deficiency, there were only small differences in the frequency of AF$^+$ cells, AF signal intensity, and LAMP1 levels in microglia from 70 day old *Mbp*$^{shi/shi}$ mice as compared to control animals (*Figure 6A,B*). Importantly, however, microglia from *Mbp*$^{shi/shi}$ mice upregulated markers known to be induced by the phagocytosis of apoptotic neuronal bodies such as CLEC7A/DECTIN1 (*Krasemann et al., 2017*; *Figure 6—figure supplement 1A,B*). Altogether, these observations validated that myelin instability was associated with increased microglia phagocytic activities in this model while also indicating that phagocytosis of unstable myelin sheaths or cellular debris was unlikely to be the primary driver of AF accumulation with aging.

To explore whether other phagocytic activities were contributors to AF accumulation, we examined Fc-receptor mediated and Triggering Receptor Expressed on Myeloid cells 2 (TREM2)-mediated phagocytosis using *Fcer1g*$^{-/-}$ mice, which lack the common Fc-gamma chain required for functional expression of activating Fc-receptors (*Takai et al., 1994*), and *Trem2*$^{-/-}$ mice, which were reported to have microglia defective in the engulfment of apoptotic neurons, myelin debris, and synapses (*Cantoni et al., 2015*; *Hsieh et al., 2009*; *Wang et al., 2015*). Microglia isolated from 9-month-old *Fcer1g*$^{-/-}$ and 6-month-old *Trem2*$^{-/-}$ mice displayed no differences in the frequency of AF$^+$ microglia or AF intensity (*Figure 6C,E*), nor did they differ in LAMP1 levels when compared to controls (*Figure 6D,F*). Together, these results indicated that the generation of microglia AF was not dependent on either Fc-receptor- or TREM2-mediated phagocytosis.

## Disruption of autophagy and lysosomal processing alter AF accumulation within AF$^+$ microglia

Given that the proteomic analysis identified upregulation of autophagic proteins in AF$^+$ microglia, we reasoned that the recycling of cellular components and their subsequent lysosomal degradation might contribute to the accumulation of AF material in microglia. To disrupt the formation of autophagic vacuoles in microglia, *Atg5*$^{flox/flox}$ mice were crossed to the *Cx3cr1*$^{CRE-ERT2}$ mouse line and treated with tamoxifen at 6 weeks of age to induce *Atg5* conditional deletion in CX3CR1$^+$ cells (*Figure 6—figure supplement 2A*), hereafter referred to as *Atg5*$^{-/-}$. In 12-month-old *Atg5*$^{-/-}$ mice, there was a significant decrease both in the frequency and the relative AF intensity of the AF$^+$ subset (*Figure 6G*). Interestingly, in *Atg5*$^{-/-}$ mice the distribution of AF intensity in AF$^+$ cells broadened significantly compared to controls. To better characterize this change in distribution, we divided the AF$^+$ subset into AF$^{dim}$ and AF$^{hi}$ populations (*Figure 6G*). In *Atg5*$^{-/-}$ mice, the proportion of AF$^{hi}$ microglia was decreased compared to *Atg5*$^{+/+}$ (30% versus 36%, p=0.012, *Figure 6H*). While no difference was observed in LAMP1 (*Figure 6I*), CD68 was significantly lower in AF$^+$ microglia from *Atg5*$^{-/-}$ mice (*Figure 6—figure supplement 2B*).

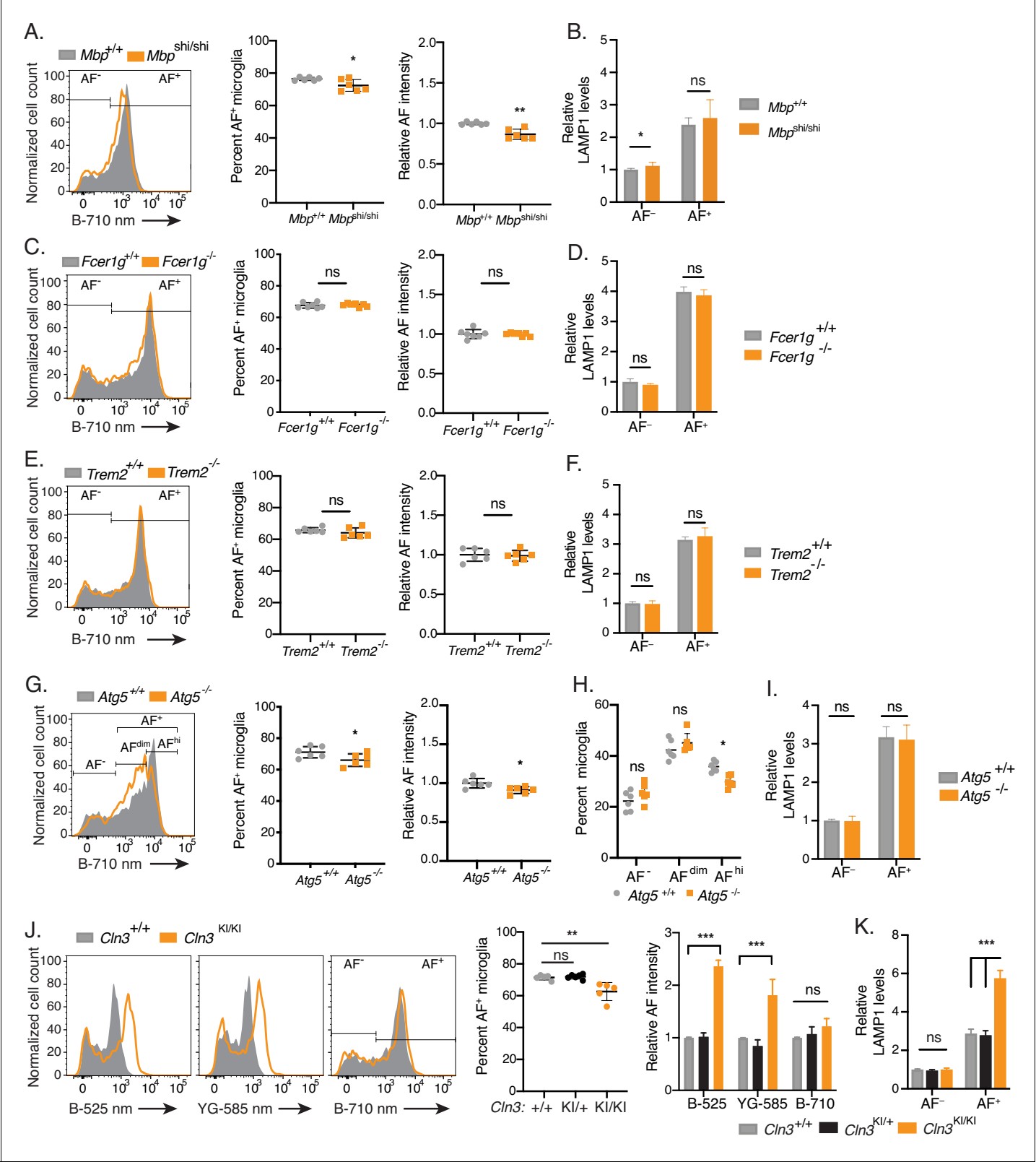

**Figure 6.** The accumulation of AF storage bodies in microglia is genetically-controlled. (A, C, E, G, H, J) Representative flow cytometry histograms and quantitation of AF subset frequency and/or AF signal intensity in AF+ microglia isolated from mice of indicated genotypes. Significance established with unpaired Student's t-test (A, C, E, G), 2-way repeated measures ANOVA followed by Sidak's post-hoc (H), and 1-way ANOVA followed by Tukey's post-hoc (J). (B, D, F, I, K) Quantitation of LAMP1 levels in microglia isolated from mice of indicated genotypes, calculated as net geometric mean

*Figure 6 continued on next page*

*Figure 6 continued*

fluorescence intensity after subtraction of background AF signal and normalization to AF intensity detected in AF⁻ microglia from wild-type age-matched control mice. Significance established with unpaired Student's t-test (B, D, F, I) and 1-way ANOVA followed by Tukey's post-hoc (K). n ≥ 6 animals per genotype group from at least two independent experiments. All groups were age-matched and all data presented as mean ± SD. AF = autofluorescence; B = blue laser; YG = yellow green laser; ns = not significant, *p<0.05, **p<0.01, ***p<0.001. See also *Figure 6—figure supplements 1* and *2*.

The online version of this article includes the following figure supplement(s) for figure 6:

**Figure supplement 1.** Characterization of microglia isolated from *Mbp*-deficient mice.
**Figure supplement 2.** Characterization of microglia isolated from *Atg5*-deficient mice.

We next examined the contribution of lysosomal pathways in the regulation of AF accumulation. To this end, we isolated microglia from *Cln3*$^{\Delta ex7/8}$ homozygous knock-in mice (hereafter referred as *Cln3*$^{KI/KI}$), which express the most common loss-of-function allelic variant of the lysosomal gene *Cln3* found in patients with juvenile-forms of neuronal ceroid lipofuscinosis (NCL) (*Cotman et al., 2002*). While the proportion of AF⁺ microglia was modestly decreased in 5-month-old *Cln3*$^{KI/KI}$ mice, AF⁺ microglia isolated from *Cln3*$^{KI/KI}$ animals showed a striking increase in AF signal intensity compared to microglia from control mice (*Figure 6J*). Furthermore, the AF signal intensity was not equivalently increased across the entire light emission spectrum, with the highest difference (2.4-fold on average) observed in the short emission wavelengths (525 nm) whereas differences were more limited at 585 nm and no longer detected at 710 nm (*Figure 6J*), likely reflecting differences in both the magnitude and composition of AF accumulation in *Cln3*$^{KI/KI}$ animals. Consistent with a lysosomal origin of AF signal in wild-type mice (*Figure 3E,F*), LAMP1 protein was increased by more than 2-fold in AF⁺ microglia from *Cln3*$^{KI/KI}$ mice (*Figure 6K*). In contrast, LAMP1 levels in AF⁻ microglia remained unchanged across all genotypes (*Figure 6K*), establishing that lysosomal dysfunction selectively affected the subset of microglia accumulating AF material. In summary, autophagy and lysosomal dysfunction respectively attenuated and increased the accumulation of AF material in AF⁺ microglia, thereby establishing autophagy and lysosomal biology as important mechanisms contributing to the formation of the AF⁺ microglia subset.

## Advanced aging and lysosomal dysfunction lead to the preferential cellular loss of the AF$^{hi}$ microglia population

Given that the relative proportion of AF⁺ microglia remained steady during the first year of life (*Figure 2E*), we next analyzed AF microglia subsets during advanced aging, defined in mice as 18 to 24 months of age. Similar to what we observed between 3 and 12 months of age (*Figure 2C*), the maximal intensity of AF in AF⁺ cells continued to increase between 12 and 24 months of age (*Figure 7A*). In contrast to the largely bimodal distribution of AF seen during the first 12 months of age, the distribution observed at 24 months of age was more continuous, with a noticeable downward shift in AF intensity detected on a per cell basis (*Figure 7A*). To quantify this change in distribution, we divided the AF⁺ subset into AF$^{dim}$ and AF$^{hi}$ populations as done previously (*Figure 6G*). While at 12 months of age 50% of microglia were AF$^{hi}$, this population decreased to 32% on average by 24 months of age, with a concomitant increase in AF$^{dim}$ and AF⁻ cells (*Figure 7B*). This apparent relative loss of the AF$^{hi}$ microglia subset was accompanied by a 31% decrease in the overall number of microglia recovered from 24-month-old mice as compared to 12-month-old animals. This decreased recovery did not affect all AF microglia subsets equally but was uniquely attributed to a 58% decrease in the absolute number of AF$^{hi}$ cells (*Figure 7C*).

To explore the possible factors involved in the preferential loss of AF$^{hi}$ microglia with age, we examined rates of proliferation and cell death in adult and aged animals. The frequency of microglia positive for AnnexinV increased in aged mice (4.0 ± 0.9%, n = 8) as compared to adult mice (1.9 ± 0.2%, n = 7) and was mostly attributed to AF$^{hi}$ and AF$^{dim}$ cells, which showed a 2-fold increase in the cell death rate at 24 months of age as compared to 12 months of age (*Figure 7D,E*). By contrast, the cell death rate seen in AF⁻ cells did not significantly increase between 12 and 24 months of age. When quantifying specifically early-apoptotic (AnnexinV⁺ DAPI⁻) cells, AF$^{hi}$ and AF$^{dim}$ microglia exhibited similar relative increases in early-apoptotic rates between 12 and 24 months of age (*Figure 7—figure supplement 1A*), altogether suggesting that increased rates in apoptotic and necrotic

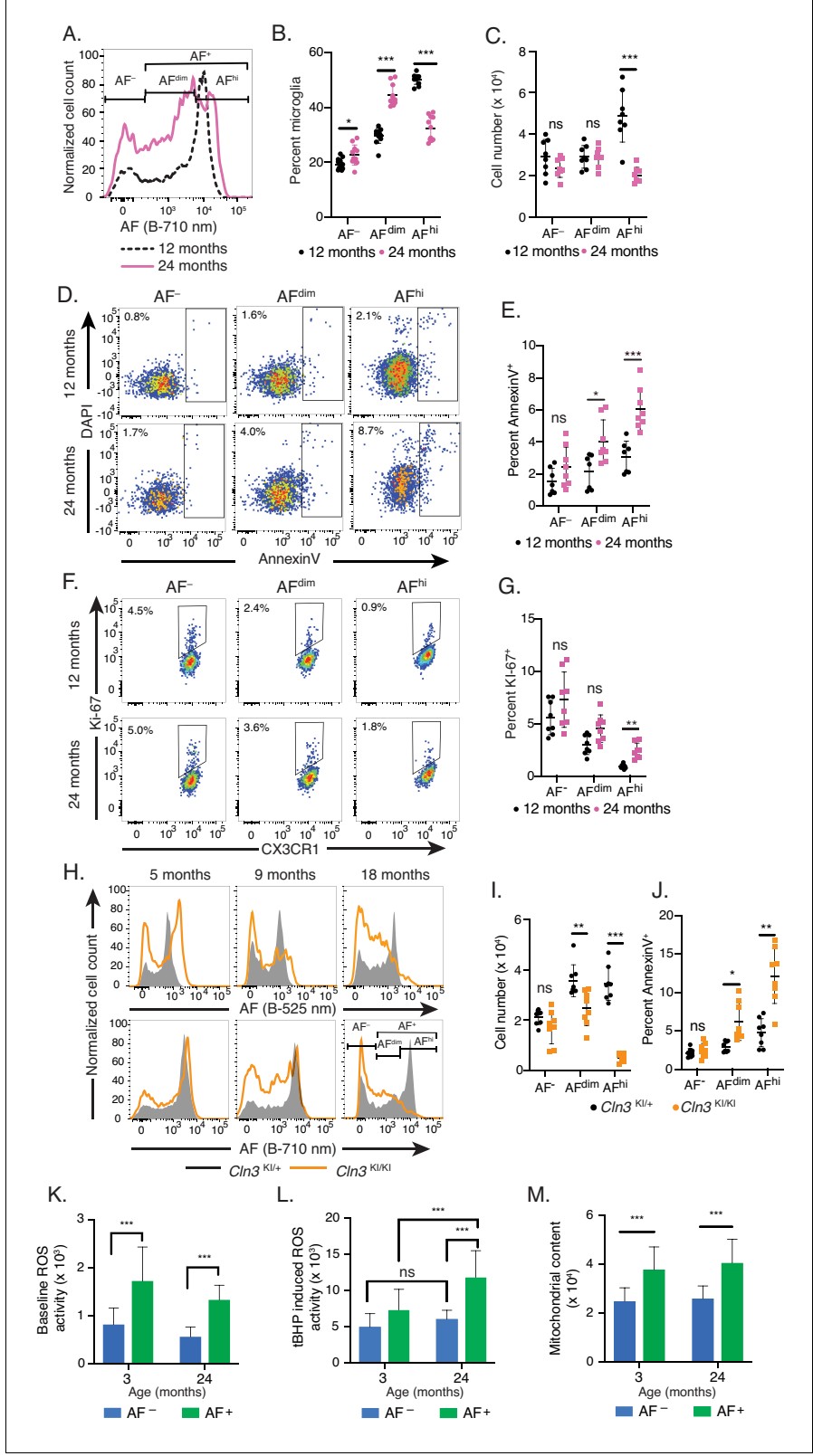

**Figure 7.** Advanced aging and lysosomal dysfunction lead to the preferential cellular loss of the AF^hi microglia population. (**A**) Representative flow cytometry histograms and (**B, C**) corresponding quantitation of AF subset frequencies (**B**) and absolute cell numbers (**C**) in 12- and 24-month-old naïve mice. Significance established with 2-way repeated measures ANOVA followed by Sidak's post-hoc. (**D–G**) Representative scatterplots depicting DAPI

*Figure 7 continued on next page*

*Figure 7 continued*

and AnnexinV staining (**D**) and KI67 and CX3CR1 staining (**F**) in microglia subsets from 12- and 24-month-old mice and corresponding quantitation of the frequency of AnnexinV$^+$ (**E**) and KI67$^+$ (**G**) cells in each microglia subset. Significance established with 2-way repeated measures ANOVA followed by Sidak's post-hoc. (**H**) Representative flow cytometry histograms of AF intensity in microglia from mice of indicated genotypes and ages and detected using two distinct cytometer channels and (**I**) corresponding quantitation of absolute cell numbers of microglia subsets in 18-month-old animals. Significance established with 2-way ANOVA followed by Sidak's post-hoc. (**J**) Quantitation of percent microglia positive for AnnexinV in indicated AF subsets from 18-month-old animals of indicated genotypes. (**K–L**) Quantitation of CellROX DeepRed ROS-indicator staining in microglia subsets from mice at indicated ages and after subtraction of background AF signal at (**K**) baseline and (**L**) after treatment with ROS inducer tBHP. (**M**) Quantitation of Mitotracker dye staining as an indicator of mitochondrial content in microglia subsets from mice at indicated ages. Significance established with 2-way ANOVA followed by Sidak's post-hoc. $n \geq 7$ animals per genotype group from at least two independent experiments. All data presented as mean ± SD. AF = autofluorescence; B = blue laser; ROS = reactive oxygen species; tBHP = tert Butyl hydroperoxide; ns = not significant *$p<0.05$ **$p<0.01$ ***$p<0.001$. See also *Figure 7—figure supplement 1*. The online version of this article includes the following figure supplement(s) for figure 7:

**Figure supplement 1.** Lysosomal dysfunction leads to the preferential cellular loss of the AF$^{hi}$ microglia population.

death were both contributing to the overall 2-fold higher cell death rate seen in AF$^{hi}$ and AF$^{dim}$ microglia with aging. Unlike cell death rates, homeostatic proliferation rates at 12 months of age were generally higher in AF$^-$ microglia than in AF$^{dim}$ (by 2-fold on average) and AF$^{hi}$ cells (by 7-fold on average) and did not change with aging, with the exception of AF$^{hi}$ microglia which showed a 2.6-fold increase in proliferative rate between 12- and 24 months of age (*Figure 7F,G*).

Because microglia from *Cln3*$^{KI/KI}$ mice accumulated higher levels of AF in the AF$^+$ population (*Figure 6J*), we reasoned that accelerated accumulation of AF material may exacerbate the differences in AF subset survival. Accordingly, as early as at 9 months of age and more prominently at 18 months of age, we observed a marked decrease in the frequency of the AF$^{hi}$ subset, using channels that either showed increased AF signal at 5 months of age (e.g., Blue-525 nm) or channels that showed no increase in AF signal at this age (e.g., Blue-710 nm) (*Figure 7H* and *Figure 7—figure supplement 1B,C*). This decreased proportion also correlated with AF dose-dependent reductions in microglia cell numbers (by 30% and 85%, respectively, for AF$^{dim}$ and AF$^{hi}$ microglia, *Figure 7I*) and increases in overall cell death and early-apoptotic rates in AF$^{hi}$ and AF$^{dim}$ microglia, but not AF$^-$ cells from *Cln3*$^{KI/KI}$ mice as compared to control mice (by 3.0- and 2.0-fold, respectively, *Figure 7J* and *Figure 7—figure supplement 1D*). Altogether, these observations supported a model where the age-associated progressive accumulation of AF material directly caused increased cell death rates, thereby leading to the selective decline of the AF$^+$ subset observed during natural aging and the premature collapse of this subset upon accelerated accumulation of AF caused by genetic manipulation of the *Cln3* pathway.

## The AF$^+$ subset is associated with an increased production of reactive oxygen species

Based on proteomic analyses, we hypothesized that the higher apoptotic rates observed in AF$^+$ cells could result from mitochondrial dysfunction, the highly catabolic metabolism of AF$^+$ cells or their reliance on fatty acid β-oxidation as a source of energy, all metabolic processes which are known to generate high levels of ROS. As measured by a ROS-sensitive cellular dye, AF$^+$ microglia from 3- and 24-month-old mice displayed baseline ROS levels that were on average 2-fold and 2.4-fold higher than those observed in AF$^-$ microglia, respectively (*Figure 7K*). After treatment with tert-Butyl hydroperoxide (tBHP), a potent inducer of cellular ROS, AF$^+$ microglia generated ROS levels that were 1.5-fold and 2-fold higher than those observed in AF$^-$ microglia at 3 and 24 months of age, respectively (*Figure 7L*). tBHP-induced ROS generation increased with aging and this increase was selectively observed in AF$^+$ microglia that generated 1.6-fold more ROS at 24 months than at 3 months of age (*Figure 7L*).

To determine whether differences in mitochondrial content were contributing to the increased levels of ROS observed in AF$^+$ cells, we assayed microglia cellular content using the Mitotracker

probe. AF$^+$ microglia exhibited, 1.5-fold and 1.6-fold higher levels of Mitotracker fluorescence on average than AF$^-$ cells at 3 and 24 months of age, respectively (*Figure 7M*). There was no age-dependent increase in Mitotracker signal detected for AF$^+$ microglia. Together, these data established that AF accumulation in AF$^+$ microglia was associated with increased generation of ROS, which may contribute to the changes selectively observed in AF$^+$ physiology with aging and lysosomal dysfunction.

## Discussion

Generally, a cellular subset should exhibit shared properties with other cells within the cell type while exhibiting unique features and selective physiological functions that are independent of their microenvironment or external stimuli. It has therefore been suggested that true microglial subsets should be defined in unchallenged conditions by their intrinsic properties which translate into unique physiological functions during development or aging (*Stratoulias et al., 2019*). This report describes novel microglia subsets that precisely satisfy each criteria of this definition: (1) Cellular autofluorescence constitutes a cell-intrinsic photophysical parameter that unexpectedly distributed in a bimodal fashion in microglia; (2) The distinct subsets of microglia identified by this parameter were found in healthy, non-challenged conditions in both mice and non-human primates, irrespective of brain region or local microenvironment (*e.g.*, various levels of myelin content such as in Shiverer mice) and independently of any specific stimulus or age (*e.g.*, these subsets were present in young healthy brain as well as in aged animals); (3) AF$^+$ cells selectively accumulated lysosomal storage bodies, therefore identifying the subset of microglial cells endowed with clearance functions in the brain and suggesting that this intrinsic AF property reflected unique physiological functions; (4) These subsets were detected longitudinally at a fixed ratio throughout most of adult life, implying the existence of homeostatic mechanisms maintaining a strict abundance ratio between them. Microglia that were negative for autofluorescence were detected even in advanced aging conditions, excluding the possibility that these subsets could be simple cellular phenotypes. While DAM, MgnD, LDAM and PAM microglia states (*Hagemeyer et al., 2017*; *Kamphuis et al., 2016*; *Keren-Shaul et al., 2017*; *Krasemann et al., 2017*; *Li et al., 2019*; *Marschallinger et al., 2020*) were previously described in disease settings, aging and during development, functionally-defined populations of microglia have not been identified or characterized in steady-state settings – to the best of our knowledge –, despite the heterogeneity of microglia observed by mass cytometry and single cell transcriptomic methods (*Hammond et al., 2019*; *Li et al., 2019*; *Masuda et al., 2019*; *Matcovitch-Natan et al., 2016*; *Mrdjen et al., 2018*). The novel subsets described here, which we propose to name AF$^+$ and AF$^-$, are predicted to have distinct functions and potentially represent the first defined subsets of microglia in the unchallenged healthy brain.

Lines of evidence point to a model where AF$^+$ microglia derive from the gradual and synchronous conversion of a subset of AF$^-$ cells: (1) AF was found to linearly increase from 3 months to 12 months of age, indicative of a progressive and cumulative biological process during aging; (2) Microglia proliferation following depletion was largely restricted to the AF$^-$ subset which fully replenished within 14 days; (3) The post-depletion re-emergence of the AF$^+$ subset was delayed until after proliferation had subsided. One limitation of these repopulation experiments is that we were unable to exclude the possibility that a small number of surviving AF$^+$ cells were responsible for repopulating the AF$^+$ niche over a longer period of time. Only future lineage tracing experiments or the adoptive-transfer of congenically-labeled AF$^+$ and AF$^-$ microglia into neonatal hosts as recently described (*Hasselmann et al., 2019*) will definitely establish the ontogenic relationship between these microglia subsets. Our repopulation results are in agreement with the reported decrease in both lipofuscin content and CD68$^+$ lysosomal volume described by O'Neil and colleagues at 21 days following PLX5622-induced microglia depletion and repopulation (*O'Neil et al., 2018*). While the authors concluded that enforced microglia turnover reversed age-related lipofuscin accumulation, our data indicate that a discrete AF$^+$ microglia population re-emerges post depletion and that repopulating AF$^+$ microglia progressively accumulate AF with time, further establishing that AF accumulation is a progressive physiological process directly linked to aging. However, several important questions remain to be addressed in future studies, among which are: (1) the formal demonstration of the irreversibility of AF accumulation with aging; (2) the identification of the molecular cues responsible for the

conversion from AF$^-$ to AF$^+$; and (3) the elucidation of the mechanisms maintaining AF subset ratios.

Historically, the gradual increase of AF in the brain with aging has been attributed to the accumulation of lipofuscin within postmitotic neurons and glial cells (*Nakanishi and Wu, 2009*; *Seehafer and Pearce, 2006*; *Xu et al., 2008*). Lipofuscin is characterized by a distinctive cytoplasmic accumulation of non-degradable fluorescent storage material comprised of highly crosslinked, polymeric and oxidized macromolecules as well as metal cations (*Brunk and Terman, 2002*). While lipofuscin typically displays spectral excitation and emission maxima in the range of 320–480 nm and 460–630 nm, respectively (*Jung et al., 2010*; *Moreno-García et al., 2018*; *Warburton et al., 2007*), the exact composition is known to vary between brain regions and cell types, likely impacting its precise spectral properties (*Gilissen and Staneva-Dobrovski, 2013*). In contrast, several exciting laser wavelengths (405, 488, 561, 640 nm) led to the emission of AF in microglia while the AF emission spectrum was also characterized by an equally wide range of wavelengths (450–780 nm) observed on a per cell basis. The latter property suggests that the AF signal detected within each AF$^+$ microglia originates from complex subcellular content that may only partially overlap with traditional lipofuscin detected in other cell types. For example, while ceroid and lipofuscin exhibit some overlapping characteristics, these storage materials have unique compositions that in turn display subtle spectral differences (*Seehafer and Pearce, 2006*). The possibility that microglia AF observed by flow cytometry is detecting a wide range of storage material is further supported by the genotype-dependent spectral differences observed in *Cln3*$^{KI/KI}$ microglia, which displayed increased fluorescence intensity in select channel wavelengths, likely reflecting the specific accumulation of ceroid and not all possible subtypes of autofluorescence-emitting storage materials that may accumulate in AF$^+$ microglia with normal aging. One limitation of this study is that the identification of AF subsets currently requires analysis of a single cell suspension by flow cytometry. Efforts leveraging proteomics-generated DEPs are currently ongoing to understand the spatial distribution in situ of AF$^+$ and AF$^-$ microglia in the context of different CNS regions and in relation to other cell types or subcellular structures such as synapses or apoptotic bodies. Understanding how aging and lysosomal dysfunction modify the spatial distribution of these two microglia subsets within the CNS should ultimately improve our understanding of the differential origin and composition of storage bodies accumulating in AF$^+$ microglia. When imaged by TEM, several types of storage compartments were detected in AF$^+$ microglia, among which were: lipid-based storage material which was reminiscent of the lipid droplets recently described in lipid droplet-accumulating microglia (LDAM) (*Marschallinger et al., 2020*; crystalline deposits which resembled cholesterol crystals previously described in foamy macrophages, lipid laden microglia and endothelial cells (*Baumer et al., 2017*; *Cantuti-Castelvetri et al., 2018*; *Klinkner et al., 1995*; *Tangirala et al., 1994*); and finally, curvilinear and fingerprint-like pattern depositions, which are hallmark features of storage disorders such as NCL and progranulin haploinsufficiency (*Anderson et al., 2006*; *Cotman et al., 2002*; *Ward et al., 2017*) and suggested a mechanistic convergence between lipofuscinosis/ceroid pathologies and microglia dysfunction associated with aging (*Spittau, 2017*).

In addition to differences in subcellular content and population dynamics, AF$^+$ microglia showed distinct differences at the proteome level, characterized by increased representation of endolysosomal and autophagic pathways as well as proteins involved in oxidative phosphorylation, catabolic metabolism and fatty acid β-oxidation. These proteomic differences are reminiscent of previously reported effects of aging on the microglia proteome, which consisted of a bioenergetic shift towards fatty-acid utilization, perturbations in the regulation of gene expression and an upregulation of proteins suggestive of mitochondrial dysfunction (*Flowers et al., 2017*). However, while those changes were observed in microglia from 24-month-old mice, we detected these differences in AF$^+$ microglia from mice as young as 12-month-old, suggesting that this aging signature may be primarily driven by the AF$^+$ subset. GO term enrichment and IPA analyses also identified dysregulation of mTOR-related proteins and pathways in AF$^+$ microglia. mTORC1, which integrates upstream pathways sensing nutrient availability, cellular energy and growth factors, regulates cell growth through its control of metabolism, autophagy, protein translation and organelle biogenesis (*Liu and Sabatini, 2020*). The activity of mTORC1 is tightly controlled by its recruitment to the lysosomal surface by the Rag-Ragulator complex, where is it brought into close proximity to the activating GTPase Rheb. The elevated levels of Ragulator components (LAMTOR1, 2, 4, 5) and Rag members RRAGA/B and RRAGC in AF$^+$ cells are likely reflective of the increased lysosomal content present in these cells.

The oxidative and catabolic metabolic profile of AF$^+$ microglia along with their overrepresentation of autophagic proteins points to an impaired activation of mTOR in AF$^+$ microglia and suggests that pharmacological modulation of mTOR may restore AF$^+$ microglia physiology, a hypothesis that remains to be tested. Intriguingly, recent work exploring the effects of elevated mTORC1 signaling in *Tsc1*-deficient microglia described increased lysosomal biogenesis and phagocytosis that was coincident with decreased synaptic density and a more reactive microglial morphology (*Zhao et al., 2018*).

In prior studies which analyzed lipofuscin in microglia during aging (*Mendes-Jorge et al., 2009*; *Moreno-García et al., 2018*; *Nakanishi and Wu, 2009*), the assumption was that the accumulation occurred equivalently across all microglial cells. Instead, we have established that the progressive increase of AF material and associated accumulation of LAMP1$^+$ storage bodies are occurring exclusively within AF$^+$ microglia, highly suggestive of functional differences between subsets. In contrast, AF$^-$ microglia remained devoid of AF signal during natural aging and even when lysosomal function was impaired by the *Cln3*$^{\Delta ex7-8}$ mutation. The persistence of the AF$^-$ subset in those perturbed conditions suggests specific mechanisms or regional cues governing the distinct functions unique to each AF subset that will be the object of future investigation. Interestingly, aging-related myelin degradation, shedding and subsequent phagocytosis by microglia were proposed to be the primary contributors to the accumulation of lipofuscin (*Safaiyan et al., 2016*). However, our observations in the MBP-mutant Shiverer mice (*Molineaux et al., 1986*), which exhibit CNS hypomyelination and myelin sheath instability (*Bird et al., 1978*; *Kirschner and Ganser, 1980*; *Privat et al., 1979*; *Safaiyan et al., 2016*; *Weil et al., 2016*), do not fully support this model as the bimodal AF distribution was largely unaltered in *Mbp*$^{shi/shi}$ microglia despite their upregulation of CLEC7A/DECTIN1, which likely reflected their active phagocytosis of apoptotic bodies and cellular debris caused by myelin instability. Therefore, myelin phagocytosis and degradation are unlikely to be the primary contributor to the microglia AF signal. This conclusion is compatible with observations made in the subretinal layers of the eye where the increased lipofuscin signal observed in microglia with aging or age-related macular degeneration was not linked to myelin phagocytosis, but to the phagocytosis of rod outer segments and apoptotic retinal pigment epithelial cells (*Santos et al., 2010*; *Xu et al., 2008*). While the lack of alterations in AF with the loss of Fc-receptor- or TREM2-mediated phagocytosis indicates that those particular pathways are not key contributors, AF accumulation may still be impacted by alternative phagocytic pathways active in microglia that were not tested in this study, such as TAM (Tyro3, Axl, Mer) (*Fourgeaud et al., 2016*; *Tufail et al., 2017*), SIRPα-CD47 (*Hutter et al., 2019*) and C3-CD11b (*Fu et al., 2012*; *Schafer et al., 2012*). In contrast, we established that lysosomal and autophagosomal degradation pathways are primary mechanisms of accumulation of AF material, as evidenced by the accelerated and decreased AF accumulation respectively observed in AF$^+$ microglia from *Cln3*$^{\Delta ex7-8}$ mice and *Atg5*-deficient microglia. Of all cell types in the brain, microglia express the highest transcript levels of *Cln3*, which is involved in the regulation of lysosomal pH, cathepsin activity, and endocytic trafficking (*Cárcel-Trullols et al., 2017*; *Cotman and Staropoli, 2012*; *Golabek et al., 2000*; *Schmidtke et al., 2019*). Although the disruption of *Cln3* expression was present in both AF$^+$ and AF$^-$ cells in *Cln3*$^{\Delta ex7-8}$ mice, the latter subset was unaffected by the perturbation of the lysosomal pathway, suggesting that CLN3-dependent lysosomal degradation is dispensable in AF$^-$ microglia, which further highlights the molecular differences between AF$^+$ and AF$^-$ microglia.

While lipofuscin accumulation has been linked to age-related neuronal and microglial functional decline (*von Bernhardi et al., 2015*; *Brunk and Terman, 2002*; *Höhn and Grune, 2013*; *Jung et al., 2007*; *Kurz et al., 2008*; *Safaiyan et al., 2016*; *Sierra et al., 2007*), the ability to precisely characterize its functional impact was impeded by the inability to isolate microglia which accumulated AF storage material from those which did not. Here, we overcame this limitation by developing a method permitting the isolation of these microglia subsets and established that the age-dependent accumulation of AF storage material was associated with increased mitochondrial mass and ROS production, decreased homeostatic proliferation and increased cell death rates in AF$^+$ cells. Increased ROS production by aging AF$^+$ microglia could be a direct consequence of the accumulation of AF material as lipofuscin can incorporate transition metals such as copper and iron, which form a redox-active surface catalyzing the Fenton reaction and promote mitochondria-independent cytotoxic effects (*Höhn et al., 2010*). Alternatively, the increased generation of ROS in AF$^+$ could result from mitochondrial dysfunction, a pathway that was enriched in the AF$^+$ proteome or the highly catabolic

metabolism of AF$^+$ cells and their reliance on fatty acid β-oxidation which is another major a source of ROS. In addition to ROS, the accumulation of undegraded macromolecules within lysosomes, which is known to inhibit key catabolic enzymes and permeases (*Lamanna et al., 2011*; *Prinetti et al., 2011*; *Walkley and Vanier, 2009*), could further propel the lysosomal system into complete dysfunction. The subsequent overaccumulation of storage material may also physically damage the lysosomal membranes, causing dispersal of storage contents into the cytosol or the extracellular space and subsequent inflammasome activation and neuronal cell death, as it was shown recently in a genetic model for acid sphingomyelinase deficiency and in a demyelination model (*Cantuti-Castelvetri et al., 2018*; *Gabandé Rodríguez et al., 2018*). Foam cell macrophages, which are generated by the overaccumulation of oxidized low-density lipid protein and cholesterol in atherosclerotic plaques, have been shown to contribute to vascular pathology via these precise mechanisms (*Childs et al., 2016*; *Duewell et al., 2010*; *Gibson et al., 2018*; *Hakala et al., 2003*). Interestingly, foam cell macrophages exhibit many of the ultrastructural features observed in aged AF$^+$ microglia and thus may share common pathways precipitating cellular dysfunction and tissue pathology. Lastly, another possible mechanism by which AF material could trigger microglia dysfunction and cytotoxic effects is via its binding and inhibition of the proteasome and the associated perturbation of cellular proteolytic functions, as it was previously reported for lipofuscin (*Höhn et al., 2011*; *Sitte et al., 2000*; *Szweda et al., 2003*). Despite the striking stability of the AF subset ratio throughout most adult mice, cumulative and age-dependent AF accumulation ultimately resulted in the collapse of the ratio of AF$^+$ to AF$^-$ microglia in advanced aging, likely due to one of the aforementioned mechanisms. The accelerated accumulation of AF material in AF$^+$ microglia from $Cln3^{\Delta ex7-8}$ mice phenocopied the effect of advanced aging on microglia by selectively increasing apoptotic rates of AF$^+$ cells and resulting in an early and precipitous decline of the AF$^+$ subset. Altogether, these results suggest a convergence of mechanisms operating during aging and in lysosomal storage disorders whereby the accumulation of storage bodies in AF$^+$ cells drives microglia dysfunction, possibly contributing to neurodegeneration known to be associated with both conditions.

Together, these results provide novel insight into microglia physiology by identifying two previously unknown subsets which exist at steady-state in the murine and non-human primate CNS. The ability to identify and isolate distinct microglia subsets via a cell-intrinsic, label-free method presents future opportunities to further characterize any additional functions that may shed light on their subset-specific roles in maintaining the delicate balance between CNS homeostasis and neurodegeneration. The significance of these subsets is clearly highlighted by a model of juvenile neuronal ceroid lipofuscinosis in which lysosomal dysfunction was born solely by AF$^+$ microglia, whereas AF$^-$ microglia appeared unaffected. Lastly, the finding that the progressive depletion of AF$^+$ microglia occurred as function of natural brain aging raises the possibility that this subset may be functionally protective and warrants further investigation into the biological factors regulating their survival.

# Materials and methods

**Key resources table**

| Reagent type (species) or resource | Designation | Source or reference | Identifiers | Additional information |
|---|---|---|---|---|
| Chemical compound, drug | BLZ945; CSF1R inhibitor | PMID:24056773 | CAS No. 953769-46-5 | |
| Chemical compound, drug | Tamoxifen | Millipore-Sigma | Cat# T5648 | |
| Biological sample (*C. macaque*) | NHP brain samples | Charles River Labs | | freshly isolated same day |
| Other | DAPI | ThermoFisher | Cat# 62248 | Cell impermeant blue-fluorescent DNA stain for viability used at a concentration of 0.1 µg/mL |
| Antibody | anti-mouse CD11b BV510 (rat monoclonal) | Biolegend | RRID:AB_2561390 | FlowCyt (1:200) |

*Continued on next page*

*Continued*

| Reagent type (species) or resource | Designation | Source or reference | Identifiers | Additional information |
|---|---|---|---|---|
| Antibody | anti-mouse CD45 BV785 (rat monoclonal) | Biolegend | RRID:AB_2564590 | FlowCyt (1:200) |
| Antibody | anti-mouse CX3CR1 BV421 (rat monoclonal) | Biolegend | RRID:AB_2565706 | FlowCyt (1:200) |
| Antibody | anti-mouse P2RY12 PE (rat monoclonal) | Biolegend | RRID:AB_2721644 | FlowCyt (1:200) |
| Antibody | anti-mouse TMEM119 (rabbit monoclonal) | AbCam | RRID:AB_2744673 | FlowCyt (1:100) |
| Antibody | anti-rabbit AlexaFluor 488 (rat monoclonal) | Biolegend | RRID:AB_2563203 | FlowCyt (1:400) |
| Other | AnnexinV AlexaFluor647 | ThermoFisher | Cat# A23204 | Apoptotic cell labeling reagent |
| Antibody | anti-mouse CD68 FITC (rat monoclonal) | Biolegend | RRID:AB_10575475 | FlowCyt (1:300) |
| Antibody | anti-mouse LAMP1 Alexa Fluor647 (rat monoclonal) | Biolegend | RRID:AB_571990 | FlowCyt (1:200) |
| Antibody | anti-mouse KI-67 BV421 (rat monoclonal) | Biolegend | RRID:AB_2629748 | FlowCyt (1:100) |
| Antibody | anti-mouse Ly6G PE (rat monoclonal) | BD Biosciences | RRID:AB_394208 | FlowCyt (1:100) |
| Antibody | anti-mouse Ly6C PE (rat monoclonal) | Biolegend | RRID:AB_1186133 | FlowCyt (1:100) |
| Antibody | anti-mouse CD3 PE (rat monoclonal) | BD Biosciences | RRID:AB_395699 | FlowCyt (1:100) |
| Antibody | anti-mouse CD19 PE (rat monoclonal) | BD Biosciences | RRID:AB_395050 | FlowCyt (1:100) |
| Antibody | anti-mouse NK1.1 PE (rat monoclonal) | BD Biosciences | RRID:AB_396674 | FlowCyt (1:100) |
| Antibody | anti-mouse CD45 AF488 (rat monoclonal) | Biolegend | RRID:AB_493531 | FlowCyt (1:100) |
| Antibody | anti-mouse CX3CR1 BV785 (rat monoclonal) | Biolegend | RRID:AB_2565938 | FlowCyt (1:200) |
| Antibody | anti-mouse CD11b APC (rat monoclonal) | Biolegend | RRID:AB_312794 | FlowCyt (1:200) |
| Antibody | anti-NHP CD45 BV786 (mouse monoclonal) | BD Biosciences | RRID:AB_2738454 | FlowCyt (5 µL/test) |
| Antibody | anti-NHP CD11b BV510 (mouse monoclonal) | BD Biosciences | RRID:AB_2737996 | FlowCyt (5 µL/test) |
| Genetic reagent (*M. musculus*) | *Cx3cr1tm2.1(cre/ERT2) Jung* (Cx3cr1–CreERT2) | PMID:23273845 | RRID:IMSR_JAX:020940 | Dr. Steffen Jung (Weizmann Institute of Science, Israel) |
| Genetic reagent (*M. musculus*) | *Atg5*flox | PMID:16625204 | | Dr. Noboru Mizushima, (University of Tokyo) |
| Genetic reagent (*M. musculus*) | *Fcer1g*-/- | Taconic Biosciences | Model 583 | PMID:8313472 |
| Genetic reagent (*M. musculus*) | *Trem2*-/- | KOMP repository | RRID:MMRRC_050209-UCD | |
| Genetic reagent (*M. musculus*) | *Mbp*shi | Jackson Laboratory | RRID:IMSR_JAX:001428 | PMID:6168677 |
| Genetic reagent (*M. musculus*) | *Cln3e × 7/ex8* (Cln3 KI) | PMID:12374761 | | Dr. Susan Cotman (Harvard Medical School) |

*Continued on next page*

*Continued*

| Reagent type (species) or resource | Designation | Source or reference | Identifiers | Additional information |
|---|---|---|---|---|
| Software, algorithm | FlowJo V10 | TreeStar | RRID:SCR_008520 | |
| Software, algorithm | Prism 8.0 | Graphpad | RRID:SCR_002798 | |
| Software, algorithm | FIJI | FIJI | RRID:SCR_002285 | |
| Software, algorithm | MaxQuant Version 1.6.0.16 | Max Planck | RRID:SCR_014485 | |
| Software, algorithm | Perseus | Max Planck | RRID:SCR_015753 | |
| Software, algorithm | Ingenuity Pathway Analysis | Qiagen | RRID:SCR_008653 | |
| Other | Percoll | GE Healthcare | Cat# 17-0891-01 | Medium for density gradient centrifugation |
| Other | Mitotracker DeepRedFM | ThermoFisher | Cat# M22426 | Mitochondrial fluorescent dye, used at 50 nM |
| Other | CellROX DeepRed | ThermoFisher | Cat# C10491 | ROS fluorescent detection agent, used at 500 nM |
| Sequence based reagent | *Atg5* Taqman primers | ThermoFisher | AssayID: Mm01187301_m1, Cat# 4351372 | Spanning exons 3–4 |
| Commercial assay or kit | RNEasy Plus Micro | Qiagen | Cat# 74034 | |
| Commercial assay or kit | SuperScript IV VILO kit | ThermoFisher | Cat# 11754050 | |
| Sequence based reagent | Gapdh endogenous control primers, *M. musculus* | ThermoFisher | Cat# 4352339E | |
| Commercial assay or kit | QuantiTect Multiplex PCR | Qiagen | Cat# 204541 | |
| Software, algorithm | PANTHER | PANTHER | RRID:SCR_004869 | |
| Peptide, recombinant protein | Lysyl Endopeptidase, Mass Spectrometry Grade (Lys-C) | Wako | Cat# 121–05063 | |
| Peptide, recombinant protein | Sequencing Grade Modified Trypsin | Promega | Cat# V5113 | |

## Mice

This study was performed in accordance with the National Institutes of Health Guide for the Care and Use of Laboratory Animals. Research animals at Biogen were housed under specific pathogen free conditions in an AAALAC accredited facility with a 12 hr - 12 hr light - dark cycle and environmental conditions controlled at 21°C and 40–60% humidity. Animals handled according to an approved institutional animal care and use committee (IACUC) protocol (#756). This study was reviewed and approved by the Massachusetts General Hospital (MGH) Subcommittee of Research Animal Care (SRAC), which serves as the Institutional Animal Care and Use Committee (IACUC) for MGH (Protocol #2008N000013). C57BL/6J, B6.129P2-*Apoe*$^{tm1Unc}$/J, C3Fe.SWV-*Mbp*$^{shi}$/J (*Chernoff, 1981*), B6J.B6N(Cg)-*Cx3cr1*$^{tm1.1(cre)Jung}$/J (*Cx3cr1*$^{CreERT2}$) (*Yona et al., 2013*) were purchased from Jackson Labs. B6.129S-*Atg5*$^{<tm1Myok>}$ (*Atg5*$^{flox}$) (*Hara et al., 2006*) were acquired from Dr. Noboru Mizushima (University of Tokyo). *Cln3*$^{\Delta ex7/ex8}$ knock-in mice (*Cotman et al., 2002*) were generously provided by Dr. Susan Cotman (Harvard Medical School) and carry a ~ 1 kb genomic deletion in the lysosomal gene *Cln3* that is orthologous to the commonly observed mutation in Juvenile Neuronal Ceroid Lipofuscinosis patients. This exonic deletion results in the production of an alternatively spliced mRNA species encoding a detectable non-truncated CLN3 mutant protein. B6.129P2-*Fcer1g*$^{tm1Rav}$ N12 (*Fcer1g*$^{-/-}$) (*Takai et al., 1994*) were purchased from Taconic. Trem2$^{tm1}$$^{(KOMP)VLcg}$ (*Trem2*$^{-/-}$) were acquired from the KOMP repository. *Atg5*$^{-/-}$ mice were generated by

crossing the $Atg5^{flox}$ and $Cx3cr1^{CreERT2}$ lines. Age-matched cohorts were used in all experiments. All mice used in studies were female unless otherwise noted. Mice were euthanized by CO2 inhalation.

## Tamoxifen treatment

One gram of tamoxifen (Cat# T5648, Millipore-Sigma) was dissolved in 5 mL ethanol and then mixed with 45 mL of corn oil USP (Cat# CO136, Spectrum Chemical) warmed at 37°C for a final concentration of 20 mg/mL. 6 week old mice were given two 200 µL subcutaneous injections of tamoxifen-corn oil, spaced 48 hr apart to induce genetic recombination.

## Isolation of murine microglia for flow cytometry

All solutions were used ice-cold unless otherwise noted. Immediately following $CO_2$ euthanasia, mice were transcardially perfused with 20 mL of phosphate buffered saline (PBS) containing 3 mM EDTA. Brains were dissected and transferred into a 15 mL conical tube filled with Hanks' Balanced Salt Solution (HBSS) on ice. For tissue homogenization, brains were transferred to a Petri dish and minced over ice with a scalpel. Minced tissue was transferred to a 7 mL glass dounce homogenizer (Cat# 57542, Wheaton) containing 5 mL of HBSS + 25 mM HEPES and homogenized with a pestle for approximately 10 strokes each. The single cell suspension was transferred to a 15 mL conical tube and centrifuged at 300xg for 5 min at 4°C. The supernatant was aspirated, and the cell pellet was gently resuspended in 1 mL of fetal bovine serum (FBS). A 33% isotonic Percoll solution was prepared by combining 9 mL of Percoll (Cat# 17-0891-01, GE Healthcare) with 1 mL of 10X HBSS and 20 mL of 1X HBSS + 25 mM HEPES. 9 mL of the 33% isotonic Percoll solution was then added and the cell suspension mixed by inversion. 1 mL of a 10% FBS/HBSS solution was overlaid on the Percoll suspension and the cells were centrifuged at 800 g for 15 min at 4°C with no brake. The resulting cell pellet was washed once in 10 mL HBSS + 25 mM HEPES, centrifuged at 300xg for 5 min at 4°C and resuspended in a final volume of 1 mL HBSS + 25 mM HEPES. Antibody staining was performed in a 96 well v-bottom plate (Cat# 3897, Corning) with the following antibodies after Fc-receptor blocking for 5 min (anti-CD16/32, clone 93, Biolegend): anti-CD45 BV785 (clone 30-F11, BioLegend), anti-CD11B BV510 (clone M1/70, BioLegend), anti-CX3CR1 BV421 (clone SA011F11, BioLegend), and anti-P2RY12 PE (clone S16007D, BioLegend). For unconjugated anti-TMEM119 (clone 101–6, Abcam) an anti-rabbit AlexaFluor488 secondary antibody (clone Poly4064, BioLegend) was used. DAPI (Cat# 62248, ThermoFisher Scientific) was used to assess viability. For apoptotic cell analysis, AnnexinV conjugated to AlexaFluor647 (Cat# A23204 ThermoFisher Scientific) was used according to the supplier protocol after surface staining. For intracellular antigens, cell suspensions were fixed and permeabilized using the Transcription Factor Staining kit (Cat# 00-5523-00, ThermoFisher Scientific) after surface staining with ZombieUV Fixable Viability dye (Cat# 423107 BioLegend). Fixation was followed by staining with CD68 FITC (clone FA11, BioLegend) LAMP1 AlexaFluor647 (clone 1D4B, BioLegend) and KI-67 BV421 (clone 11F6, BioLegend). Absolute cell counts were determined using 123count eBeads (Cat# 01-1234-42, ThermoFisher Scientific) Flow cytometry data were acquired on a five laser-equipped LSRFortessa X-20 (BD Biosciences) and analyzed in FlowJoV10 (Treestar). Non-autofluorescent peripheral immune cells were used as negative controls to determine the initial bifurcating gate to identify $AF^-$ and $AF^+$ microglia subsets. To subdivide the $AF^+$ subset, a gate encompassing $AF^{hi}$ microglia was drawn to capture the $AF^+$ gaussian peak. The $AF^{dim}$ gate was drawn between the edge of the $AF^-$ population and the base of the $AF^+$ gaussian peak (lower boundary of $AF^{hi}$ gate). Imaging flow cytometry data were acquired with an AMNIS Imagestream MKII (Millipore-Sigma) and analyzed with AMNIS IDEAS software.

## Isolation of non-human primate microglia for flow cytometry

Immediately following euthanasia, Cynomolgus macaques were transcardially perfused with an ice-cold Lactated Ringers Solution (LRS). Brains were removed and placed into ice-cold LRS and kept chilled until processing. Approximately 500 mg of tissue from the frontal cortex was excised, minced and gently homogenized on ice in a 7 mL glass dounce homogenizer (Cat# 57542, Wheaton) containing 5 mL of HBSS + 25 mM HEPES. Microglia were then isolated using the same procedure as described above for murine microglia. Antibody staining was performed in a 96 well v-bottom plate (Cat# 3897, Corning) with the following antibodies after Fc-receptor blocking (Human TruStain FcX,

Cat#422301, Biolegend): anti-CD45 BV786 (clone D058-1283, BD Biosciences), anti-CD11B BV510 (clone ICRF44, BD Biosciences).

## Isolation of murine microglia for fluorescence activated cell sorting (FACS)

Single cell suspensions were prepared as previously described for flow cytometry microglia isolations through the initial centrifugation. The resulting cell pellet was resuspended in 5 mL of a room-temperature (RT) 70% isotonic Percoll solution, which was carefully overlaid with 5 mL of a 37% RT isotonic Percoll solution. The Percoll cell suspension was then centrifuged at 800 g for 25 min at 22°C with acceleration set to 50% and brake set to 10%. Enriched microglia were then carefully collected from the 37%/70% Percoll interphase to a new 15 mL conical tube previously blocked with 2% bovine serum albumin (BSA) in PBS at RT for 3 hr. Microglia were washed once with 10 mL of ice-cold HBSS + 25 mM HEPES and centrifuged at 300 g for 5 min at 4°C. The resulting microglia cell pellet was resuspended in 100 μL of HBSS + 2% BSA. Cell suspensions were Fc-blocked and then stained with anti-CD45 AF488 (clone 30-F11, BioLegend), anti-CD11B APC (clone M1/70, BioLegend), anti-CX3CR1 BV785 (clone SA011F11, BioLegend). DAPI was used for viability and a dump gate in the PE channel was created with the following antibodies: anti-Ly6G PE (clone 1A8, BD Biosciences), anti-Ly6C PE (clone HK1.4, BioLegend), anti-CD3 PE (clone 17A2, BD Biosciences), anti-CD19 PE (clone 1D3, BD Biosciences), anti-NK1.1 PE (clone PK136, BD Biosciences). Cell sorting was performed on a BD FACSAria Fusion instrument equipped with an 85 μm nozzle.

## Proteomics sample preparation and data acquisition

75,000 microglia were FACS-isolated into tubes containing the dried residual of 150 μL 8M urea, 5 mM EDTA, 0.1M Tris/HCl pH 8.5, resulting in final volumes of 300 μL. The suspensions were mixed and stored frozen until used. Lysates were thawed and treated with a E220 focused beam ultrasonicator (Covaris) 20 times at 2 s each with 150W output. Lysates were clarified by centrifugation, reduced with 10 mM final concentration of DTT for 20 min and subsequently alkylated with 30 mM 2-iodoacetamide for 30 min at room temperature. After a two-fold dilution in water, samples were digested overnight with 0.05 μg LysC (Wako) and 0.2 μg modified trypsin (Promega). Peptides were acidified to a final concentration 0.5% (v/v) trifluoroacetic acid and desalted using C18 StageTips (*Rappsilber et al., 2007*). 50% of resulting peptides were separated on a 50 cm, 75 μm inner diameter EasySpray column packed with 2 μm PepMap C18 RSLC material (ThermoFisher) over 120 min using an EASY-nLC 1200 system (ThermoFisher). Peptides were analyzed online with an Orbitrap Fusion-Lumos mass spectrometer (ThermoFisher) in data-dependent acquisition mode. Full scans were acquired at a resolution of 240,000 in the Orbitrap analyzer and the most abundant precursors were selected in a 1 s scan cycle for higher-energy dissociation (HCD) with a 0.7 Th isolation window, a target of 10,000 ions and a maximum injection time of 25 ms, followed by detection fragment ion in the ion trap.

## Proteomics data analysis

Raw data were processed in MaxQuant Version 1.6.0.16 (*Cox and Mann, 2008*) and searched with Andromeda (*Cox et al., 2011*) against a comprehensive SwissProt release for mouse with a false discovery rate of 1% at the peptide and protein level. Identifications of MS1 features were transferred between samples using the Match Between Runs option with a 0.2 min matching tolerance. Label-free protein quantification was performed using the MaxLFQ algorithm (*Cox et al., 2014*). Data analysis was performed in Perseus (*Tyanova et al., 2016*). After log transformation and applying a filter for proteins quantified in at least 50% of samples in either group, missing values were imputed with a normal distribution with a mean shifted down by 1.8-fold and a width of 0.3-fold in relation to the standard distribution of measured values of the respective sample. *p* values were calculated using Student's t-test and the false discovery rate was controlled with the Benjamini-Hochberg method. Proteins were determined to be differentially expressed if they met an adjusted *p* value threshold of <0.01 and an absolute fold change difference threshold of >1.3. Human orthologs of identified mouse proteins were assigned using Panther and GO term enrichment of differentially expressed proteins was calculated with the Panther Overpresentation Test (*Mi et al., 2019*). Canonical pathway

analysis and identification of upstream regulators was performed with Ingenuity Pathway Analysis (Qiagen).

## Transmission electron microscopy

Microglia from pooled animals (n = 10 per age) were FACS-isolated directly into tubes containing 4% glutaraldehyde/0.1M sodium-cacodylate solution, pH 7.4 (Electron Microscopy Sciences, Hatfield, PA). Fixed cells were pelleted and post-fixed in 1.0% osmium tetroxide in cacodylate buffer for 1 hr at RT, rinsed in cacodylate buffer, and stabilized using 2% agarose in PBS. Agarose-embedded pellets were dehydrated through a graded series of ethanols to 100%, then dehydrated briefly in 100% propylene oxide. Specimens were infiltrated in a mix of 2:1 propylene oxide/Eponate resin (Ted Pella, Redding, CA) 2 hr at room temperature on a gentle rotator, then switched into a 1:1 mix of propylene oxide/Eponate resin and allowed to infiltrate overnight on a gentle rotator. 24 hr later, specimens were placed into fresh 100% Eponate resin and allowed to infiltrate for several hours, then embedded in flat molds with fresh 100% Eponate. Polymerization occurred within 24–48 hr at 60˚C. Thin (70 nm) sections were cut using a Leica EM UC7 ultramicrotome, collected onto formvar-coated grids, contrast-stained with uranyl acetate and Reynold's lead citrate and examined in a JEOL JEM 1011 transmission electron microscope at 80 kV. Images were collected using an AMT digital imaging system with proprietary image capture software (Advanced Microscopy Techniques, Danvers, MA). TEM image analysis and area determinations for subcellular features were calculated in FIJI (*Schindelin et al., 2012*). Cytosolic area values were calculated by determining the full cellular area and subtracting the nuclear region.

## Mitochondrial content

Mitotracker DeepRedFM (Cat# M22426, ThermoFisher Scientific) reagent was used according to the supplier protocol. Briefly, a single cell suspension of isolated microglia was surface stained with CD45 BV786 and CD11B BV510. Cells were pelleted and then resuspended gently in 500 µL pre-warmed 37˚C staining solution containing 50 nM of Mitotracker reagent. Cells were incubated for 30 min at 37˚C in a humidified 5% $CO_2$ incubator then pelleted and resuspended in 200 µL of HBSS +2% BSA for analysis by flow cytometry.

## ROS detection

CellROX Deep Red (Cat# C10491, ThermoFisher Scientific) was used according to supplier protocol. Briefly, a single cell suspension of isolated microglia was surface stained with CD45 BV786 and CD11B BV510. Cells were pelleted and then resuspended gently in prewarmed RPMI1640 (Cat# 11875085, ThermoFisher Scientific) containing 10% FBS. Cells were treated with a 100 µM solution containing tert-butyl hydroperoxide for 30 min at 37˚C in a humidified 5% CO2 incubator to induce ROS. CellROX reagent was then added to a final concentration of 500 nM and the cells were incubated for 45 min at 37˚C. Cells were pelleted, resuspended in 200 µL of prewarmed RPMI1640/10% FBS containing DAPI (0.1 µg/mL final concentration) and then promptly analyzed on a flow cytometer.

## Microglia depletion

To deplete microglia, 14-month-old mice were orally treated q.d. for 7 days with BLZ945 at a dose of 200 mg/kg body-weight (*Krauser et al., 2015*) in a 20% captisol solution. Vehicle-treated mice received only 20% captisol solution. Tissues were collected at the indicated time after last dose.

## RNA isolation and quantitative real-time PCR

50,000–100,000 microglia were FACS-isolated directly into chilled RNA/DNA free microfuge tubes containing 500 µL RLT-plus lysis buffer supplemented with 1% 2-mercaptoethanol. RNA was isolated with the RNEasy Plus Micro Kit (Cat# 74034, Qiagen) according to the kit protocol. RNA quality was assessed using the Agilent RNA6000 Pico kit (Cat# 5067–1513, Agilent) on an Agilent Bioanalyzer system. RNA quantity was determined with the Quant-iT RiboGreen Fluorescence assay (Cat# R11490, ThermoFisher Scientific). Isolated RNA was then converted to cDNA using the SuperScript IV VILO kit (Cat# 11754050, ThermoFisher Scientific) per manufacturer protocol. Microglia cDNA was combined with a QuantiTect Multiplex PCR (Cat# 204541 Qiagen) master mix containing

Taqman primers spanning *Atg5* exons 3–4 (AssayID Mm01187301_m1, Cat# 4351372, ThermoFisher Scientific) and *Gapdh* endogenous control primers (Cat# 4352339E, ThermoFisher Scientific). qRT-PCR reaction was run for 40 cycles on a QuantStudio12K Flex instrument (ThermoFisher Scientific) and relative quantity (RQ) of *Atg5* transcript was calculated using the $2^{\Delta\Delta Ct}$ method.

### Statistical analysis

Statistical analysis was performed using Prism software (GraphPad, San Diego, CA). To determine significance values when comparing AF subsets, a paired, two-tailed t-test or 2-way repeated measures ANOVA with Tukey's or Sidak's post-hoc multiple comparison's test were utilized. For comparisons between genotypes, significance testing was performed using two-tailed independent t-test or 1-way ANOVA followed by Tukey's post-hoc for multiple comparisons. For BLZ945 experiments, a 1-way ANOVA followed by Dunnett's post-hoc test for multiple comparisons was used. A value of $p<0.05$ was considered statistically significant. All data were presented as the mean with standard deviation.

## Acknowledgements

The authors thank M Klein, A Nowell and Dr. E Butz for assistance in collection of $Cln3^{\Delta ex7/8}$ mice and the Batten Disease Research Foundation for funding and support of Dr. S Cotman's laboratory. The authors acknowledge the support of the Boston University Biomolecular Pharmacology training program. The authors also acknowledge D Capen and the Microscopy Core of the Center for Systems Biology/Program in Membrane Biology (Mass General Hospital), which is partially supported by an Inflammatory Bowel Disease Grant DK043351 and a Boston Area Diabetes and Endocrinology Research Center (BADERC) Award DK05752.

## Additional information

### Competing interests

Jeremy Carlos Burns, Bunny Cotleur, Dirk M Walther, Bekim Bajrami, Stephen J Rubino, Ru Wei, Nathalie Franchimont, Michael Mingueneau: currently a full-time employee of Biogen. Richard M Ransohoff: currently a full-time employee of Third Rock Ventures. The other author declares that no competing interests exist.

### Funding

| Funder | Author |
| --- | --- |
| Batten Disease Research | Susan L Cotman |

The funders had no role in study design, data collection and interpretation, or the decision to submit the work for publication.

### Author contributions

Jeremy Carlos Burns, Conceptualization, Formal analysis, Investigation, Visualization, Methodology, Writing - original draft; Bunny Cotleur, Investigation; Dirk M Walther, Data curation, Formal analysis, Investigation, Methodology; Bekim Bajrami, Formal analysis, Investigation, Methodology; Stephen J Rubino, Investigation, Writing - review and editing; Ru Wei, Supervision, Methodology, Writing - review and editing; Nathalie Franchimont, Supervision, Writing - review and editing; Susan L Cotman, Conceptualization, Resources, Supervision; Richard M Ransohoff, Michael Mingueneau, Conceptualization, Supervision, Writing - review and editing

### Author ORCIDs

Jeremy Carlos Burns https://orcid.org/0000-0003-3853-0237
Bunny Cotleur https://orcid.org/0000-0003-4571-6435
Bekim Bajrami https://orcid.org/0000-0002-4129-9390
Ru Wei https://orcid.org/0000-0002-7646-9046

Nathalie Franchimont (iD) https://orcid.org/0000-0002-2265-8378
Susan L Cotman (iD) https://orcid.org/0000-0003-3114-0543
Richard M Ransohoff (iD) https://orcid.org/0000-0003-0175-6910
Michael Mingueneau (iD) https://orcid.org/0000-0002-3873-7329

## Ethics

Animal experimentation: This study was performed in accordance with the National Institutes of Health Guide for the Care and Use of Laboratory Animals. Research animals at Biogen were housed in an AAALAC accredited facility and handled according to an approved institutional animal care and use committee (IACUC) protocol (#756). This study was reviewed and approved by the Massachusetts General Hospital (MGH) Subcommittee of Research Animal Care (SRAC), which serves as the Institutional Animal Care and Use Committee (IACUC) for MGH (Protocol #2008N000013).

## Decision letter and Author response

Decision letter https://doi.org/10.7554/eLife.57495.sa1
Author response https://doi.org/10.7554/eLife.57495.sa2

## Additional files

### Supplementary files

• Transparent reporting form

### Data availability

Data are available via ProteomeXchange with identifier PXD017505. Submission details: Project Name: Autofluorescence positive and negative microglia constitute novel subsets found in healthy brain. Project accession: PXD017505.

The following dataset was generated:

| Author(s) | Year | Dataset title | Dataset URL | Database and Identifier |
|---|---|---|---|---|
| Burns JC, Cotleur B, Walther DM, Bajrami B, Rubino SJ, Wei R, Franchimont N, Cotman SL, Ransohoff RM, Mingueneau M | 2020 | Autofluorescence positive and negative microglia constitute novel subsets found in healthy brain | http://proteomecentral. proteomexchange.org/ cgi/GetDataset?ID= PXD017505 | ProteomeXchange, PXD017505 |

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
