## [Decision Letter]

**Acceptance summary:**

This study provides a novel description of microglial heterogeneity during the process of aging, with careful characterization of a helpful and easily accessible biomarker-the presence or absence of autofluorescence. Increased autofluorescence in AF+ microglia occurs during normal aging, can be regulated by lysosomal pathways and autophagy, and leads to increased microglial apoptosis and ROS production.

**Decision letter after peer review:**

Thank you for submitting your article "Storage body accumulating microglia, a novel subset in healthy brain selectively impacted by aging" for consideration by *eLife*. Your article has been reviewed by Suzanne Pfeffer as the Senior Editor and Reviewing Editor and three reviewers. The reviewers have opted to remain anonymous.

The reviewers have discussed the reviews with one another and the Reviewing Editor has drafted this decision to help you prepare a revised submission. While we normally highlight the specific actions needed to revise the story, I instead include the detailed reviewer comments as many of the comments relate to the presentation of the data rather than serving as a list of needed experiments. Because there is no functional characterization, this would either need to be completed, or comments pertaining to function would need to be amended and a section outlining these caveats and future steps included. Also, data for other brain regions which was alluded to should be included; integration with the body of literature on lipofuscin would also be helpful.

We hope you will find the comments constructive in preparing a revised manuscript that we believe should be considered in the category of RESOURCE article.

We would like to draw your attention to changes in our revision policy that we have made in response to COVID-19 (https://elifesciences.org/articles/57162). Specifically, when editors judge that a submitted work as a whole belongs in *eLife* but that some conclusions require a modest amount of additional new data, as they may with your paper, we are asking that the manuscript be revised to either limit claims to those supported by data in hand, or to explicitly state that the relevant conclusions require additional supporting data.

Our expectation is that the authors will eventually carry out the additional experiments and report on how they affect the relevant conclusions either in a preprint on bioRxiv or medRxiv, or if appropriate, as a Research Advance in *eLife*, either of which would be linked to the original paper.*Reviewer #1:*

In "Storage body accumulating microglia, a novel subset in healthy brain selectively impacted by aging", the authors identify two subsets of microglia based on the presence or absence of autofluorescence (termed AF+ and AF-). The authors used young and aged mice to characterize changes in the AF+ microglial population during normal aging, including in knockout mice for phagocytic and lysosomal pathway genes. The authors conclude that (1) increased intensity of autofluorescence in AF+ microglia occurs during normal aging and is associated with increased storage body accumulation and proteomic changes, (2) AF intensity can be regulated by lysosomal pathways and autophagy, and (3) in advanced aging the accumulation of AF leads to increased microglial apoptosis and ROS production.

This study provides a novel description of microglial heterogeneity during the process of aging, with careful characterization of a helpful and easily accessible biomarker. It is an appealing topic to neuroimmunology and aging fields. Although the manuscript does not delineate the mechanisms regulating autofluorescence, nor its consequences on physiology, these topics represent exciting future directions. There are several points that, if addressed, would broaden appeal, increase interpretability, and/or strengthen conclusions.

1) All presented data examines single cell suspensions of microglia. Identification of AF+ microglia requires flow cytometry, the specific mechanisms regulating microglial autofluorescence are not known, and AF+ microglia have a range of different cellular inclusions. It is therefore challenging to know how to identify AF+ versus AF- microglia in tissue sections, but this would greatly enhance our ability to gauge their relevance. If the authors have any data or thoughts about how to identify AF+ microglia in situ, they would be highly useful to the field. If not, it could be helpful to mention this understandable limitation.

2) Subsection “Cellular autofluorescence identifies two discrete microglia subsets”: "AF microglia subsets were detected across different regions of the brain when isolated from cerebellum, cortex and hippocampus (data not shown)." Given ongoing debates around regional microglial heterogeneity, this is an interesting and important statement. If data are already in hand pertaining to regional differences in the frequency of AF+ and AF- microglia (or lack thereof), this would be of great appeal. If not, this sentence could be removed, as it is currently provocative but difficult to interpret.

3) Subsection “Microglia AF subsets exhibit differential population dynamics upon depletion and replenishment of the microglia niche”: "Because AF+ microglia were virtually absent during this early repopulation phase (Figure 4B), these proliferation kinetics implied that the AF- subset of microglia was the predominant subset responsible for the repopulation of the microglia compartment following depletion and that AF+ microglia were derived from the conversion over time of a subset of AF- microglia." The authors lack direct evidence that AF- cells progress into AF+ cells, and it is possible that a small number of remaining AF+ microglia can directly repopulate the AF+ niche, just with slower proliferation kinetics. New experimental data is not needed, but alternative hypotheses should be discussed.

4) Regarding Figure 7: To identify true, early apoptotic cells DAPI-AnnexinV+ cells should be the only ones counted. Cells stained for both AnnexinV and DAPI are not necessarily apoptotic, since membrane permeability permits Annexin V binding to the inner membrane leaflet. Cells may have a permeable membrane due to other pathways of cell death i.e. necroptosis, necrosis, pyroptosis. Presenting data on only DAPI-AnnexinV+ cells would increase interpretability of these results and strengthen the conclusion that AFhigh microglia are more prone to apoptotic cell death. Alternatively, the authors could change their language to include the possibility of other types of cell death occurring in AF+ microglia, or note these limitations in the Discussion section.*Reviewer #2:*

This manuscript describes a new classification system for microglia in the rodent and non-human primate brain that is based on autofluorescence. These AF+ and AF- microglia are equally distributed across brain regions (though this is not shown and would be a nice addition as the data is stated as being generated), does not differ according to sex, and maintains similar percentages following repopulation after depletion of microglia. Unlike other classifications of microglia in the disease state (e.g. disease-associated microglia, DAMs), AF+/AF- cells are present across species, and the proteomic analysis that suggests that have lysosomal instead of phagocytic differences is very interesting (this was validated using Trem2-/- and Fcer1g-/- mice). Overall, this is an interesting hypothesis, and provides a new avenue for characterizing microglia for future functional investigation.

My concerns are based largely on this function characterization – as while the authors state they have provided an improvement on other recently published transcriptomic studies by assessing function of these AF+/AF- microglia, they have not. What they have provided is a proteomic analysis of these cells – as no functional assays are provided in the manuscript. This does not detract from the overall hypothesis of the manuscript, but it would benefit from some taming of the language.

As there has been much controversy in the microglia field about binary nomenclature, perhaps some caution here is prudent. If the cells can be shown to have a functional difference, this would go a long way to dispel concerns in the field (otherwise more appropriate descriptions and removal of statements that suggest functional testing has been completed here would suffice).

The authors state that they have measured similar AF+/AF- cells in the cerebellum, cortex, and hippocampus. Given this statement is provided as proof of continuity of the subsets at several places throughout the manuscript – these data should be shown.

The data presented in Figure 1C is great, and the percentages seem more or less correct for the B-710 group of cells, but appears more 50/50 for R-780 cells. These are presumably lowly abundant? Can some quantification of these subtypes be extracted from these data?

The authors spend considerable time describing morphological changes in Figure 3 (panels A) – suggesting that AF- cells are more regularly shaped that AF+ cells, however the representative images in Figure 3A appears to show a different story – with AF- microglia having many 'ruffled' edges, while AF+ microglia are smooth. This is particularly noticeable in the 18 month cells. This highlights a real problem in the field of describing cell subsets using morphology – it can be misleading and biased. Considering the clear AF phenotypes, the morphology of the cells is not a strong addition here. Differences in cellular inclusions remain clear, though this might only be true at 3 months. This remains a weak section of the manuscript (which on the whole is strong and exciting). Similarly, in panels B the only described component here that is clearly visible is the crystal inclusions (white arrows). The curvilinear and fingerprint like profiles are not visible. This may be a resolution problem in review images, but this should be carefully checked and amended if necessary. This is also highlighted at the bottom of Page 16 where it is suggested that these 'crystal deposits resemble cholesterol crystals found in atherosclerotic plaues'. This is highly interesting, but this statement should be qualified or measured/validated in some way. Perhaps some careful review and rewriting could address this.

The AF+ dim/lo section of the manuscript is the weakest section of the manuscript and is not well-defined or incorporated into the remainder of the very compelling story. The dim/hi populations are not defined, and it feels like a move back to CD45hi/low populations that have caused problems in the field before. Indeed, the split of these cells in Figure 6, panel G, is arbitrary and very unclear how this was chosen. In Figure 7 dim and hi populations only appear visible on FACS plots in the 24 month animals, but the split in younger animals is still unclear, and seems arbitrary to gate them as such at earlier ages as the AF is a smooth transition from lower to higher levels*Reviewer #3:*

This manuscript unveils the presence of two distinct microglial populations defined by the presence or absence of autofluorescent (AF) material. Surprisingly, the proportion of AF+ and AF- cells is constant across animals, species (mouse and a non-human primate), and ages, while AF intensity grows linearly with aging in AF+ cells. In AF+ cells, the autofluorescence localizes in intracellular organelles. Ultrastructurally, AF+ cells exhibited large and complex storage bodies whose size markedly increases with aging, along with the signals of LAMP1 and CD68 (lysosomal membrane proteins) in AF+ (but not AF-) cells. Proteomics experiments revealed upregulation of proteins of the autophagy-lysosome pathway in AF+ cells.

The authors also show that AF+ and AF- cells exhibit dramatically different dynamics in post-microglia-depletion brain repopulation, with AF- cells repopulating the niche first, and then partially converting to AF+ cells.

Analysis of mouse lines with defective genes for myelination and phagocytosis indicated that these two pathways do not contribute to the generation of AF+ microglia. Conversely, KO of an autophagy gene decreased, whereas KO of a lysosomal gene increased, autofluorescence in AF+ microglia, revealing intersections with these pathways. Advanced aging was accompanied by a striking and selective loss of microglia with highest AF levels, which was accelerated in a genetic model of lysosomal dysfunction and was paralleled by an increase in ROS production and mitochondrial content.

The story is novel, well developed, and significant. The experiments are well controlled and largely supports the main conclusions of the work. The identification of a subpopulation of microglial cells that is likely responsible for managing catabolic challenges in aging and neurodegenerative disease is significant because it defines potential targets of treatments. More generally, the finding that microglial cells are divided in two distinct populations of fairly constant proportions is intriguing and raises several questions, including (1) How the conversion from AF- to AF+ cells is trigged at the molecular level? (2) Is this conversion irreversible? (3) What keeps the proportion of the two populations constant? These questions, however, are better addressed in follow-up studies, and I am mentioning them here only to underline how this work raises innovative questions in the field.

[Editors' note: further revisions were suggested prior to acceptance, as described below.]

Thank you for resubmitting your work entitled "Storage body accumulating microglia, a novel subset detected in the healthy brain and selectively impacted by aging" for further consideration by *eLife*. Your revised article has been evaluated by Suzanne Pfeffer (Senior Editor) and a Reviewing Editor. We would be pleased to publish this article after 2 minor issues are addressed.

Please address:

1) Figure 6—figure supplement 2, panel A lacks annotation of statistical significance testing, which should be added.

2) The title was (and remains) difficult to interpret. Please try to improve?

Reviewer #2 comments included for the authors' benefit (no changes required):

I particularly appreciate the addition of AF+ microglia data from additional brain regions. I think this adds a great deal to the manuscript to show these cells as a population that exists throughout the CNS. I look forward to the efforts described in response to reviewer #1 to visualize these cells using in situ. The high mag/high resolution EM image is really beautiful – and I appreciate this addition. I also appreciate the difficulties in isolation and culture of microglia to run functional tests. Having said this, the assays in the paper remain as descriptive and correlative, which is not a problem at all, but they remain not being functional assays. I do however appreciate that the LAMP1 staining is nice, the presence of oligo and neuron material inside the cells also suggests phagocytosis. Repopulation following depletion does provide information about a possible transition from AF- to AF+ with age, but as discussion by reviewer #1 there remains a possibility that AF+ cells repopulate from a very small population. I think however this has been addressed by changing wording – e.g. subsection “Proteomic analysis of isolated AF^+^ and AF^−^ microglia subsets reveals molecular differences in endolysosomal, autophagic and metabolic pathways” – and this is fine. As the first paper describing this subset/phenomenon, I am sure future studies will cover these specifics. The clarification on B-170 R-780 cell percentages is interesting, and the description of the laser integrity in pulling out individual populations is also interesting, but as the authors suggest likely not of biological importance.

---

## [Author Response]

Reviewer #1:In "Storage body accumulating microglia, a novel subset in healthy brain selectively impacted by aging", the authors identify two subsets of microglia based on the presence or absence of autofluorescence (termed AF+ and AF-). The authors used young and aged mice to characterize changes in the AF+ microglial population during normal aging, including in knockout mice for phagocytic and lysosomal pathway genes. The authors conclude that (1) increased intensity of autofluorescence in AF+ microglia occurs during normal aging and is associated with increased storage body accumulation and proteomic changes, (2) AF intensity can be regulated by lysosomal pathways and autophagy, and (3) in advanced aging the accumulation of AF leads to increased microglial apoptosis and ROS production.This study provides a novel description of microglial heterogeneity during the process of aging, with careful characterization of a helpful and easily accessible biomarker. It is an appealing topic to neuroimmunology and aging fields. Although the manuscript does not delineate the mechanisms regulating autofluorescence, nor its consequences on physiology, these topics represent exciting future directions.

The authors would like to thank reviewer #1 for this in-depth review and appreciation of our findings as well as for highlighting the novelty and utility of microglia autofluorescence as a biomarker of microglia heterogeneity during aging. While we agree that we did not identify the exact biochemical nature of the storage bodies associated with the AF signal nor the functional consequences of AF accumulation on microglia physiology at the organismal level, we would like to emphasize two findings which partially answer these limitations:

a) Genetic analyses shown in Figure 6 identified autophagy and lysosomal functions as key mechanisms regulating autofluorescence accumulation.

b) Characterization of the effect of advanced aging and lysosomal dysfunction on the physiology of AF^-^ and AF^+^ microglia (Figure 7) allowed us to define functional consequences of early and accelerated AF accumulation at the cellular level, including increased apoptosis, higher ROS production and selective loss of the AF^+^ cells.

There are several points that, if addressed, would broaden appeal, increase interpretability, and/or strengthen conclusions.1) All presented data examines single cell suspensions of microglia. Identification of AF+ microglia requires flow cytometry, the specific mechanisms regulating microglial autofluorescence are not known, and AF+ microglia have a range of different cellular inclusions. It is therefore challenging to know how to identify AF+ versus AF- microglia in tissue sections, but this would greatly enhance our ability to gauge their relevance. If the authors have any data or thoughts about how to identify AF+ microglia in situ, they would be highly useful to the field. If not, it could be helpful to mention this understandable limitation.

The authors fully agree with reviewer #1 that developing a microscopy method to identify AF^-^ and AF^+^ microglia subsets *in situ* would further our understanding of these microglia subsets in the context of CNS tissue regions and their interactions with other cell types or subcellular structures (e.g., synapse or apoptotic bodies). Efforts for developing IHC/IF methods to localize AF^+^ and AF^-^ microglia *in situ* are currently ongoing and leveraging our proteomic dataset to define suitable markers. We have now updated the Discussion section to highlight this limitation of our findings and ongoing efforts to develop methods to identify these subsets *in situ*.

2) Subsection “Cellular autofluorescence identifies two discrete microglia subsets”: "AF microglia subsets were detected across different regions of the brain when isolated from cerebellum, cortex and hippocampus (data not shown)." Given ongoing debates around regional microglial heterogeneity, this is an interesting and important statement. If data are already in hand pertaining to regional differences in the frequency of AF+ and AF- microglia (or lack thereof), this would be of great appeal. If not, this sentence could be removed, as it is currently provocative but difficult to interpret.

We agree with reviewer #1 that the detection of AF subsets across different anatomical regions of the brain is an important finding. We have now included the corresponding data (Figure 1—figure supplement 1C,D) which establishes the presence of both AF^-^ and AF^+^ microglia in cerebellum, cortex and hippocampus. This figure panel is now described in the Results section.

3) Subsection “Microglia AF subsets exhibit differential population dynamics upon depletion and replenishment of the microglia niche”: "Because AF+ microglia were virtually absent during this early repopulation phase (Figure 4B), these proliferation kinetics implied that the AF- subset of microglia was the predominant subset responsible for the repopulation of the microglia compartment following depletion and that AF+ microglia were derived from the conversion over time of a subset of AF- microglia." The authors lack direct evidence that AF- cells progress into AF+ cells, and it is possible that a small number of remaining AF+ microglia can directly repopulate the AF+ niche, just with slower proliferation kinetics. New experimental data is not needed, but alternative hypotheses should be discussed.

The authors agree that without additional lineage tracing experiments, it is technically possible that a small number of surviving AF^+^ cells were responsible for repopulating the AF^+^ niche over a longer period of time. We have now highlighted this alternative hypothesis in the revised version of the manuscript (Discussion section). While we can’t directly exclude this alternative hypothesis, we had included several lines of evidence that argued against that possibility (Discussion section). In the absence of lineage tracing capabilities available to us at this time to label and trace AF^+^ and AF^-^ microglia, we are currently exploring alternative strategies to tackle this question, such as the transplantation of FACS-sorted microglia AF subsets from congenic donors into neonate recipients (PMID: 31375314). This future avenue of investigation is now highlighted in the Discussion section.

4) Regarding Figure 7: To identify true, early apoptotic cells DAPI-AnnexinV+ cells should be the only ones counted. Cells stained for both AnnexinV and DAPI are not necessarily apoptotic, since membrane permeability permits Annexin V binding to the inner membrane leaflet. Cells may have a permeable membrane due to other pathways of cell death i.e. necroptosis, necrosis, pyroptosis. Presenting data on only DAPI-AnnexinV+ cells would increase interpretability of these results and strengthen the conclusion that AFhigh microglia are more prone to apoptotic cell death. Alternatively, the authors could change their language to include the possibility of other types of cell death occurring in AF+ microglia, or note these limitations in the discussion section.

We would like to thank reviewer #1 for highlighting this lack of precision and accuracy in how we presented cell death data in Figure 7. The initial data presented were indeed inclusive of multiple modes of cell death and did not strictly reflect apoptotic rates, but overall cell death rates. The text has been modified accordingly in the Results section. We also reanalyzed the data to separately quantify apoptotic (AnnexinV+, DAPI-) and necrotic cells (AnnexinV+, DAPI+). We found that AF^hi^ and AF^dim^ microglia exhibited a 2-fold increase in both apoptotic and necrotic death rates between 12 and 24 months of age (Figure 7—figure supplement 1A), altogether suggesting that both increased apoptotic and necrotic death are contributing to the higher overall cell death rates of these subsets relatively to AF^-^ microglia. In addition to Figure 7—figure supplement 1A, we performed a similar analysis in Cln3^KI/KI^ mice and these results are now presented in Figure 7—figure supplement 1D.

Reviewer #2:This manuscript describes a new classification system for microglia in the rodent and non-human primate brain that is based on autofluorescence. These AF+ and AF- microglia are equally distributed across brain regions (though this is not shown and would be a nice addition as the data is stated as being generated), does not differ according to sex, and maintains similar percentages following repopulation after depletion of microglia. Unlike other classifications of microglia in the disease state (e.g. disease-associated microglia, DAMs), AF+/AF- cells are present across species, and the proteomic analysis that suggests that have lysosomal instead of phagocytic differences is very interesting (this was validated using Trem2-/- and Fcer1g-/- mice). Overall, this is an interesting hypothesis, and provides a new avenue for characterizing microglia for future functional investigation.

The authors would like to thank reviewer #2 for their positive feedback on our work and appreciation that microglia AF subsets are present across species, something which is not true (or at least being debated) for other subsets of microglia described previously. As per reviewer #2 suggestion, we have now included in the revised manuscript the characterization of AF subsets across different anatomical regions of the brain (Figure 1—figure supplement 1C,D). This figure establishes the presence of AF^-^ and AF^+^ microglia in cerebellum, cortex and hippocampus. Description of these observations is included in subsection” Cellular autofluorescence identifies two discrete microglia subsets”.

My concerns are based largely on this function characterization – as while the authors state they have provided an improvement on other recently published transcriptomic studies by assessing function of these AF+/AF- microglia, they have not. What they have provided is a proteomic analysis of these cells – as no functional assays are provided in the manuscript. This does not detract from the overall hypothesis of the manuscript, but it would benefit from some taming of the language.As there has been much controversy in the microglia field about binary nomenclature, perhaps some caution here is prudent. If the cells can be shown to have a functional difference, this would go a long way to dispel concerns in the field (otherwise more appropriate descriptions and removal of statements that suggest functional testing has been completed here would suffice).

We are in agreement with reviewer#2 that we did not perform assays to directly assess functional differences between microglia subsets and moderated the functional claims throughout the manuscript. The principle reasons for the decision to eschew in vitro functional assays is the well-documented, rapid loss of homeostatic microglia identity that occurs upon culture on a plastic dish (PMID:28521131, PMID:30108586) in addition to the limited physiological relevance of the corresponding in vitro assays. Instead, we chose to provide several lines of indirect in vivo and ex vivo evidence in support of functional differences between AF subsets. These observations include:

- The steady, age-dependent increases in AF signal, LAMP1 levels and storage bodies observed selectively in AF^+^ microglia (Figure 1, Figure 2 and Figure 3) which are wholly reflective of a difference in the functional abilities of these microglia subsets in either the ingestion or the degradation of extracellular material. Of note, none of these readouts increased with aging in AF^-^ microglia, thus pointing to a functional divergence in the lysosomal / degradative activity of the two subsets with aging.

- Further supporting these observations, proteins that are typically associated with oligodendrocytes (MOBP, SYNE2) or neurons (SNAP25, CBLN1, CBLN4) were detected in our proteomics dataset at significantly higher levels within AF^+^ microglia than AF^-^, providing further indirect evidence of increased phagocytic capacity of AF^+^ microglia, differences in the material they ultimately ingested, or differences in the degradative capacity of these subsets. We have now included these observations in subsection “ Myelin, Fc receptor-mediated and TREM2-mediated phagocytosis are not dominant mechanisms contributing to microglia AF accumulation” to highlight the differential detection of these neuronal/oligodendrocyte proteins.

- The differential ability of AF^+^ and AF^-^ microglia to repopulate the brain upon depletion (Figure 4), pointing to another functional difference between those subsets that is also reflected in the differential levels of homeostatic proliferation, ROS production and cell death seen in AF^+^ and AF^-^ cells (Figure 7).

-The differential genetic regulation of those subsets was highlighted by the lack of impact of *Cln3* mutation on AF^-^ microglia whereas the same mutation led to early cell death and loss of AF^+^ cells, indicating that AF^+^, but not AF^-^ microglia, rely on lysosomal-related functions and indirectly demonstrating that these subsets are endowed with distinct functions.

The authors state that they have measured similar AF+/AF- cells in the cerebellum, cortex, and hippocampus. Given this statement is provided as proof of continuity of the subsets at several places throughout the manuscript – these data should be shown.

As mentioned in our response to reviewer #1, we agree that the detection of AF subsets across different anatomical regions of the brain is an important finding. As such, we have now included the data as part of Figure 1—figure supplement 1C,D.

The data presented in Figure 1C is great, and the percentages seem more or less correct for the B-710 group of cells, but appears more 50/50 for R-780 cells. These are presumably lowly abundant? Can some quantification of these subtypes be extracted from these data?

Reviewer#2’s observation is correct in that in Figure 1C where each row represents a single analyzed cell, the proportion of single cells positive for AF in the B-710 channel is roughly 70%. In contrast, this proportion is different in R-780 channel where it is roughly 50%. We believe that this difference is only apparent and is not necessarily reflecting a different subtype of AF microglia. Instead, we favor the hypothesis that these differences are more likely caused by the higher sensitivity of the B-710 channel compared to R-780. This distinct sensitivity in AF detection is very clear in Figure 1B when considering the brightness of the AF^+^ population in those two channels. We have included scatterplots below to better illustrate the differences in sensitivity between B-710 and R-780 in detecting AF^+^ microglia at different ages. Using alternate high-sensitivity channels to detect AF, such as YG-610 or YG-670, one can identify a similar proportion of AF^+^ microglia (~70%) whereas weaker channels such as R-730 and R-670 yield lower proportions similar to the R-780 channel. These important considerations have now been added in subsection “Cellular autofluorescence identifies two discrete microglia subset”.

**Author response image 1. sa2fig1:** 

The authors spend considerable time describing morphological changes in Figure 3 (panels A) – suggesting that AF- cells are more regularly shaped that AF+ cells, however the representative images in Figure 3A appears to show a different story – with AF- microglia having many 'ruffled' edges, while AF+ microglia are smooth. This is particularly noticeable in the 18 month cells. This highlights a real problem in the field of describing cell subsets using morphology – it can be misleading and biased. Considering the clear AF phenotypes, the morphology of the cells is not a strong addition here. Differences in cellular inclusions remain clear, though this might only be true at 3 months. This remains a weak section of the manuscript (which on the whole is strong and exciting). Similarly, in panels B the only described component here that is clearly visible is the crystal inclusions (white arrows). The curvilinear and fingerprint like profiles are not visible. This may be a resolution problem in review images, but this should be carefully checked and amended if necessary. This is also highlighted at the bottom of Page 16 where it is suggested that these 'crystal deposits resemble cholesterol crystals found in atherosclerotic plaues'. This is highly interesting, but this statement should be qualified or measured/validated in some way. Perhaps some careful review and rewriting could address this.

The morphological characterization in subsection “AF+ microglia selectively accumulate intracellular storage bodies with age” was referring to intracellular storage bodies, not to the entire cell. This statement has been modified to minimize ambiguity. We apologize that the curvilinear fingerprint inclusions were not visible in the initial version of the manuscript that we submitted and will verify that the resolution is sufficient upon final submission. In the meantime, we have included below a high-resolution version of the image that was included in the initial submission.

Cholesterol crystallization is a well-characterized phenomenon in lipid-laden macrophages. The TEM image we included in the manuscript matches well with previous reports of intracellular cholesterol crystals. For example, Klinkner et al., imaged macrophages incubated in vitro with oxidized-LDL by TEM (Klinkner et al., 1995) and described extensive lysosomal lipoid bodies, many in the process of conversion from the fluid to crystalline state. Fully developed crystal profiles were often enveloped by phospholipid lamellae that outlined the crystal contours and were observed in one of two distinct shapes: a short bar shape or a long thin arc. We have included a clearer description of the TEM cholesterol crystal observation in subsection “AF+ microglia selectively accumulate intracellular storage bodies with age” as well as additional references where this finding has been previously characterized by TEM are now included in the Discussion section.

The AF+ dim/lo section of the manuscript is the weakest section of the manuscript and is not well-defined or incorporated into the remainder of the very compelling story. The dim/hi populations are not defined, and it feels like a move back to CD45hi/low populations that have caused problems in the field before. Indeed, the split of these cells in Figure 6, panel G, is arbitrary and very unclear how this was chosen. In Figure 7 dim and hi populations only appear visible on FACS plots in the 24 month animals, but the split in younger animals is still unclear, and seems arbitrary to gate them as such at earlier ages as the AF is a smooth transition from lower to higher levels

We would like to thank reviewer #2 for their overall description of this manuscript as a compelling story, but also for highlighting this weaker section of the manuscript. We do agree that the introduction of AF^dim^ and AF^hi^ subsets may be confusing, especially in Figure 6. We believe however that subdividing the AF^+^ subset into AF^dim^ and AF^hi^ subpopulations is important in quantifying the change in AF distribution within the AF^+^ subset observed with advanced aging and in conditions of lysosomal or autophagy dysfunction. We have therefore made a number of changes to address reviewer#2’s comment:

- In Figure 7A, the histogram distribution of AF intensity in microglia from 24-month old mice (i.e., in conditions of advanced aging) shows a very different AF distribution compared to microglia from 12-month animals, with the clear alteration of the bimodal distribution and emergence of cells with intermediate levels of AF. In order to account for this significant change in distribution, we subdivided the AF^+^ population into 2 groups, AF^hi^ and AF^dim^ in the same manner performed in Figure 6G. Initially, non-autofluorescent peripheral immune cells were used as negative controls to determine the initial bifurcating gate to identify AF^-^ and AF^+^ microglia subsets. To subdivide the AF^+^ subset, a gate encompassing AF^hi^ microglia was drawn to capture the AF^+^ gaussian peak. The AF^dim^ gate was drawn between the edge of the AF^-^ population and the base of the AF^+^ gaussian peak (lower boundary of AF^hi^ gate). To confirm the validity of this additional gating scheme, we leveraged the *Cln3* data. While the AF^hi^ cells normally formed a sharp gaussian peak in the B-710 channel, the peak completely dissipated at 18 months in *Cln3*^KI/KI^ animals (Figure 7H, I), with only AF^-^ cells and the intermediate AF^dim^ cells remaining. The justification for this gating strategy is now included in the subsection “Mice”.

- We also amended Figure 6G to remain consistent with the neighboring panels while also introducing the subdivision of the AF^+^ population with gating used outlined in Figure 6G. We have also included a panel with the accompanying quantification of the AF^hi^ and AF^dim^ populations (Figure 6H).

Reviewer #3:This manuscript unveils the presence of two distinct microglial populations defined by the presence or absence of autofluorescent (AF) material. Surprisingly, the proportion of AF+ and AF- cells is constant across animals, species (mouse and a non-human primate), and ages, while AF intensity grows linearly with aging in AF+ cells. In AF+ cells, the autofluorescence localizes in intracellular organelles. Ultrastructurally, AF+ cells exhibited large and complex storage bodies whose size markedly increases with aging, along with the signals of LAMP1 and CD68 (lysosomal membrane proteins) in AF+ (but not AF-) cells. Proteomics experiments revealed upregulation of proteins of the autophagy-lysosome pathway in AF+ cells.The authors also show that AF+ and AF- cells exhibit dramatically different dynamics in post-microglia-depletion brain repopulation, with AF- cells repopulating the niche first, and then partially converting to AF+ cells.Analysis of mouse lines with defective genes for myelination and phagocytosis indicated that these two pathways do not contribute to the generation of AF+ microglia. Conversely, KO of an autophagy gene decreased, whereas KO of a lysosomal gene increased, autofluorescence in AF+ microglia, revealing intersections with these pathways. Advanced aging was accompanied by a striking and selective loss of microglia with highest AF levels, which was accelerated in a genetic model of lysosomal dysfunction and was paralleled by an increase in ROS production and mitochondrial content.The story is novel, well developed, and significant. The experiments are well controlled and largely supports the main conclusions of the work. The identification of a subpopulation of microglial cells that is likely responsible for managing catabolic challenges in aging and neurodegenerative disease is significant because it defines potential targets of treatments. More generally, the finding that microglial cells are divided in two distinct populations of fairly constant proportions is intriguing and raises several questions, including (1) How the conversion from AF- to AF+ cells is trigged at the molecular level? (2) Is this conversion irreversible? (3) What keeps the proportion of the two populations constant? These questions, however, are better addressed in follow-up studies, and I am mentioning them here only to underline how this work raises innovative questions in the field.

The authors would like to thank reviewer #3 for this very positive feedback on our submission and highlighting the novelty and potential of our findings to develop future therapeutics in the context of aging and neurodegeneration. We have now revised the Discussion section to include the important questions raised by reviewer #3 which we agree are of high importance for follow-up studies.

[Editors' note: further revisions were suggested prior to acceptance, as described below.]

Please address:1) Figure 6—figure supplement 2, panel A lacks annotation of statistical significance testing, which should be added.2) The title was (and remains) difficult to interpret. Please try to improve?Reviewer #2 comments included for the authors' benefit (no changes required):I particularly appreciate the addition of AF+ microglia data from additional brain regions. I think this adds a great deal to the manuscript to show these cells as a population that exists throughout the CNS. I look forward to the efforts described in response to reviewer #1 to visualize these cells using in situ. The high mag/high resolution EM image is really beautiful – and I appreciate this addition. I also appreciate the difficulties in isolation and culture of microglia to run functional tests. Having said this, the assays in the paper remain as descriptive and correlative, which is not a problem at all, but they remain not being functional assays. I do however appreciate that the LAMP1 staining is nice, the presence of oligo and neuron material inside the cells also suggests phagocytosis. Repopulation following depletion does provide information about a possible transition from AF- to AF+ with age, but as discussion by reviewer #1 there remains a possibility that AF+ cells repopulate from a very small population. I think however this has been addressed by changing wording – e.g. subsection “Proteomic analysis of isolated AF^+^ and AF^-^ microglia subsets reveals molecular differences in endolysosomal, autophagic and metabolic pathways” – and this is fine. As the first paper describing this subset/phenomenon, I am sure future studies will cover these specifics. The clarification on B-170 R-780 cell percentages is interesting, and the description of the laser integrity in pulling out individual populations is also interesting, but as the authors suggest likely not of biological importance.

We thank the editorial staff for the quick decision on our manuscript revisions. We have amended Figure 6—figure supplement 2A and accompanying legend to include the statistics for that experiment. At the suggestion of the reviewing editor, we have also changed the manuscript title to "Differential accumulation of storage bodies with aging defines discrete subsets of microglia in the healthy brain" which we believe to be simpler and more precise than the previous title.